# Oxidative Stress in NAFLD: Role of Nutrients and Food Contaminants

**DOI:** 10.3390/biom10121702

**Published:** 2020-12-21

**Authors:** Clémence Rives, Anne Fougerat, Sandrine Ellero-Simatos, Nicolas Loiseau, Hervé Guillou, Laurence Gamet-Payrastre, Walter Wahli

**Affiliations:** 1Toxalim (Research Center in Food Toxicology), Université de Toulouse, INRA, EVT, INP-Purpan, UPS, 31300 Toulouse, France; clemence.rives@inrae.fr (C.R.); anne.fougerat@inrae.fr (A.F.); sandrine.ellero-simatos@inrae.fr (S.E.-S.); nicolas.loiseau@inrae.fr (N.L.); herve.guillou@inrae.fr (H.G.); 2Lee Kong Chian School of Medicine, Nanyang Technological University Singapore, Clinical Sciences Building, 11 Mandalay Road, Singapore 308232, Singapore; 3Center for Integrative Genomics, Université de Lausanne, Le Génopode, CH-1015 Lausanne, Switzerland

**Keywords:** steatosis, non-alcoholic fatty liver disease (NAFLD), non-alcoholic steatohepatitis (NASH), food contaminant, macronutrients, micronutrients, oxidative stress, mitochondria, reactive oxygen species (ROS)

## Abstract

Non-alcoholic fatty liver disease (NAFLD) is often the hepatic expression of metabolic syndrome and its comorbidities that comprise, among others, obesity and insulin-resistance. NAFLD involves a large spectrum of clinical conditions. These range from steatosis, a benign liver disorder characterized by the accumulation of fat in hepatocytes, to non-alcoholic steatohepatitis (NASH), which is characterized by inflammation, hepatocyte damage, and liver fibrosis. NASH can further progress to cirrhosis and hepatocellular carcinoma. The etiology of NAFLD involves both genetic and environmental factors, including an unhealthy lifestyle. Of note, unhealthy eating is clearly associated with NAFLD development and progression to NASH. Both macronutrients (sugars, lipids, proteins) and micronutrients (vitamins, phytoingredients, antioxidants) affect NAFLD pathogenesis. Furthermore, some evidence indicates disruption of metabolic homeostasis by food contaminants, some of which are risk factor candidates in NAFLD. At the molecular level, several models have been proposed for the pathogenesis of NAFLD. Most importantly, oxidative stress and mitochondrial damage have been reported to be causative in NAFLD initiation and progression. The aim of this review is to provide an overview of the contribution of nutrients and food contaminants, especially pesticides, to oxidative stress and how they may influence NAFLD pathogenesis.

## 1. Introduction and Background

### 1.1. General Overview

Non-alcoholic fatty liver disease (NAFLD) frequently co-exists with metabolic syndrome and thus often is defined as the liver expression of dyslipidemia, insulin resistance, and obesity. However, it can also occur in patients with low body mass index (<25 kg/m^2^), a condition known as lean NAFLD [1,2,3]. NAFLD is characterized by fat accumulation in hepatocytes and comprises a spectrum of disease. This spectrum ranges from simple steatosis (non-alcoholic fatty liver or NAFL) to non-alcoholic steatohepatitis (NASH) with inflammation, hepatocyte ballooning, and various levels of fibrosis, associated with a significant risk for progression to cirrhosis and liver cancer (hepatocellular carcinoma (HCC)). In fact, only 20% of NAFLD patients develop NASH, and 4% of patients with NAFL and 20% of patients with NASH develop cirrhosis [4]. Over time, up to 5% of patients with NASH will experience spontaneous regression. NASH is strongly associated with the degree of hepatic fibrosis, which is predictive of the overall mortality in patients with NAFLD [4].

NAFLD prevalence is increasing among children [5,6] and has become the most common chronic liver disease in childhood and adolescence, as it is among adults. NAFLD affects approximately 25% of the global population [7], but the prevalence varies considerably by region. It is highest in the Middle East (32%) and South America (30%) and lowest in Africa (13%). In North America and Europe, the prevalence is 24%, and it is 27% in Asia [7]. As the overweight and obesity pandemic drives the development of metabolic conditions promoting NAFLD, the health and economic burdens continuously increase, also fueled by the boom in childhood obesity and an aging population [8,9].

NAFLD is defined as >55 mg triglycerides (TGs) per gram of liver or the presence of more than 5% steatotic hepatocytes without histological damage or inflammation. Most commonly, the percentage of hepatocytes presenting lipid accumulation is divided into three ranges: 5% to 33%, 34% to 66%, and >66%. Accordingly, the severity of steatosis is considered to be mild, moderate, or severe, respectively [10]. NAFL can progress to NASH that is characterized, in addition to steatosis, by ballooning degeneration of hepatocytes and lobular inflammation. The presence of perisinusoidal fibrosis is generally not considered a prerequisite for NASH diagnosis [11]. A recent study has found that some patients with NAFLD but without NASH may develop progressive fibrosis and have an increased mortality risk regardless of their baseline NASH status [12]. Furthermore, fibrosis progression is faster in lean patients with NAFLD, who can have a higher risk of developing cirrhosis or HCC compared with NAFLD patients who are overweight [13]. This pattern suggests that once the disease has progressed to NASH, obesity may not be the key factor in fibrosis progression [1], but this association remains debated and needs further investigation.

### 1.2. Development of NAFLD

Multiple factors determine NAFLD progression, which has led to the proposed “multiple hit” hypothesis [14]. These hits, which include nutritional factors and the gut microbiota, act in the context of diverse genetic backgrounds. For several years, development of NAFLD was explained by the “two hit” theory, according to which progression to NASH from “simple” steatosis or NAFL required a second “hit,” such as oxidative stress and hepatocyte damage. Today’s perception of NAFLD is that several different pathways can drive its development, resulting in a heterogeneous range of manifestations characterized by liver and serum lipidomic signatures that reflect disease development and progression. These variants allow for classification of NAFLD into different subtypes, based on the impairments of key pathways that control fatty acid (FA) homeostasis [15]. Pathogenic drivers of NAFL and NASH may arise because of the overwhelmed capacity of the liver to handle an overflow of the primary metabolic energy substrates, carbohydrates, and FAs, leading to the production of toxic lipid species [16]. The substrate overload liver injury model of NASH pathogenesis underscores the pivotal role of free FAs (FFAs) in the disease process during which toxic metabolites promote hepatocellular stress, cellular injury, and death. The activation of hepatic stellate cells and immune cells enhances the fibrogenesis, inflammation, and genomic instability that predispose to cirrhosis and HCC. There are two sources of hepatic free FFAs: lipolysis of TGs in adipose tissue with liver delivery through the bloodstream, and de novo lipogenesis in hepatocytes, which converts excess carbohydrates, especially fructose, to FAs and contributes to the accumulation of hepatic fat in NAFLD [17]. In hepatocytes, these FAs are processed by mitochondrial beta-oxidation and re-esterification to form TGs that are stored in lipid droplets or exported into the bloodstream, carried by very low-density lipoprotein (VLDL).

### 1.3. Determinants of NAFLD/NASH Development

#### 1.3.1. Genetic Component

Clinical observations have unveiled a high variability in susceptibility to NAFLD, with some patients exhibiting NASH and others having NAFL only, despite presenting with well-established risk factors. This variability has been addressed in the last decade after findings revealed an important role for differences in genetic background. A better understanding of the genetic basis of NAFLD should benefit the development of new medications to treat advanced disease [18]. Several genetic polymorphisms associated with NAFLD susceptibility implicate proteins regulating lipid and retinoid metabolism. The most well-documented polymorphism is the Patatin-like phospholipase domain–containing protein 3 (PNPLA3) isoleucine-to-methionine substitution at position 148 (rs738409C>G coding for PNPLA3 I148M). This loss-of-function I148M variant protein has already been reviewed in several publications [1,14,18,19]. *PNPLA3* encodes the patatin-like phospholipase domain-containing protein 3, a triacylglycerol lipase that mediates triacylglycerol hydrolysis in adipocytes and hepatic lipid droplets. Significant associations have been identified between PNPLA3 I148M and liver fat, NASH, fibrosis, cirrhosis, and HCC. PNPLA3 I148M accumulates on lipid droplets and evades ubiquitylation so that its degradation is reduced. It remodels liver TGs and increases hepatic retention of polyunsaturated FAs (PUFAs) [16,20]. Furthermore, PNPLA3 has a role in hepatic stellate cell activation in response to fibrogenic stimuli and in the release of retinol by these cells [21].

The next important protein variant is the transmembrane 6 superfamily member 2 (TM6SF2) guanine-to-adenine substitution at position 167 (rs58542926 E>K, coding for TM6SF2 E167K), which also results in a loss of activity. TM6SF2 plays a role in loading TGs to apolipoprotein B 100 in the secretion pathway of hepatic VLDL. Reduced TM6SF2 activity results in triacylglycerol entrapping in lipid droplets [20,22]. Therefore, TM6SF2 E167K increases hepatic TG content while decreasing lipoprotein levels in the serum. Individuals carrying the variant gene have a higher risk for NAFLD but lower risk for cardiovascular disease [23]. Consistent with this finding, knockdown of *Tm6sf2* in mice or transient TM6SF2 overexpression alters the serum lipid profiles [24].

Another gene variant is that of the membrane-bound O-acyltransferase domain-containing 7 (MBOAT7) protein that incorporates arachidonic acid into phosphatidylinositol. In hepatocytes, the rs641738 C>T *MBOAT7* variant is associated with lower protein levels and changes in phosphatidylinositol species in the plasma. It also is associated with NAFLD severity in individuals of European descent, possibly resulting from changes in the remodeling of the liver phosphatidylinositol acyl-chain [25]. In fact, this variant increases the risk for NAFLD, including inflammation and fibrosis in white patients because of toxic accumulation of PUFAs as a consequence of their altered incorporation into hepatocyte phospholipids [20,25], and HCC in individuals without cirrhosis [26].

Glucokinase regulatory protein (GCKRP) regulates de novo lipogenesis by controlling the flux of glucose into hepatocytes. A missense loss-of-function GCKR variant (rs1260326, GCKR P446L) causes hepatic fat accumulation through stimulation of glucose uptake into hepatocytes, which fuels de novo lipogenesis [27,28].

Finally, polymorphisms within 17β-hydroxysteroid dehydrogenase type 13 (HSD17B13), a protein expressed at the surface of lipid droplets with retinol dehydrogenase activity, are associated with NAFLD, NASH, fibrosis, and cirrhosis [29,30]. Of interest, the effect of the HSD17B13 variant in NAFLD is greater in carriers of the PNPLA3 I148M variant. This observation suggests that these two risk factors interact in the pathogenesis of NAFLD. Both PNPLA3 and HSD17B13 regulate retinol metabolism, which may suggest that this metabolism in hepatic stellate cells could be involved in the pathogenesis of NAFLD [31]. These protein variants resulting from gene polymorphisms illustrate clearly the genetic contribution to NAFLD susceptibility.

#### 1.3.2. Environmental Factors

In current Western societies, unhealthy diets, lack of exercise, smoking, and alcohol consumption are typical lifestyle risk factors that contribute to NAFLD development [32,33]. In particular, excessive fructose and saturated FA consumption, combined with sedentary lifestyles have contributed to the global increase of obesity and NAFLD. Increasing epidemiological and experimental evidence also suggest that exposure to some environmental contaminants, such as pesticides, participate in NAFLD development and progression.

Nutrients

The effects of NAFLD-associated gene polymorphisms, as noted above, indicate complex interactions between genetic and environmental factors in the progression of NAFLD that are not yet completely understood [34,35,36]. Lifestyle interventions and dietary factors offer benefits in patients with NAFLD without obesity [37]. A healthy diet, together with physical activity, is commonly recommended to patients with NAFLD to improve their condition. Weight loss significantly contributes to controlling disease progression. However, despite clear documentation that dietary and exercise interventions are effective, many patients fail to adhere to the recommendations for different reasons [1,38]. A recent meta-analysis investigating the relationship between the risk for NAFLD in association with different food groups (e.g., vegetables, fruits, red meat, fish, refined grains, whole grains, soft drinks) found that red meat and soft drink intake is positively associated with NAFLD [39]. The impact of specific diets on NAFLD is now well documented, such as the differences in the effects of Westernized versus Mediterranean diets [1,38]. The Western diet (WD), which is particularly rich in saturated fat and added fructose, is associated with a greater risk for and severity of NAFLD [40]. Diets characterized by a high intake of so-called fast foods and rich in saturated FAs, trans-fats, simple sugars, red and processed meats, full-fat dairy products, and soft drinks have a deleterious effect on the liver. In contrast, Mediterranean-type diets, which are low in saturated fat and cholesterol and rich in monounsaturated fats, polyunsaturated omega-3 FAs with a balanced ratio of omega-6 to omega-3 FAs, a high content of complex carbohydrates, dietary fiber, and plant-based proteins, are beneficial to the liver and associated with lower rates of NAFLD, NASH, and fibrosis [41,42,43,44]. A systematic review evaluating the effectiveness of different dietary interventions recently showed that the Mediterranean diet is effective for reducing hepatic steatosis [45]. This diet also can reduce liver fat without changes in weight and may be particularly beneficial for patients with high genetic risk for NAFLD [46,47,48]. As might be anticipated from these findings, greater adherence to the Mediterranean diet also is associated with less severe NAFLD [49].

Among macronutrients, sugars such as sucrose and high fructose corn syrup significantly increase risk for NAFLD and NASH [50]. Carbohydrates, especially fructose, stimulate de novo lipogenesis, which accounts for 26% of increased FAs within the liver [51,52,53]. Among patients with NAFLD, dietary fructose consumption, especially in sweetened beverages, is increased [54] and associated with heightened inflammation and fibrosis [55]. Consistent with this observation, very low-carbohydrate ketogenic diets decrease hepatic lipids and improve histologic parameters including steatosis, inflammation, and fibrosis in people with obesity [56,57]. Dietary lipids are involved in NAFLD but are not the main source of liver fat in NAFLD, as only 15% of hepatic lipids in NAFLD patients trace to dietary fat [17]. Several studies have shown that high intake of saturated lipids, cholesterol, simple carbohydrates, and animal proteins is a major determinant of NAFLD and contributes to disturbances in energy homeostasis (lipid and glucose metabolism) and to hepatocyte damage [58]. Of note, cholesterol is consistently elevated in human and mouse fibrotic NASH. Its mechanistic link to NASH development has been explored recently [59]. Cholesterol upregulates the transcriptional regulator TAZ (transcriptional co-activator with PDZ-binding motif) and promotes fibrotic NASH through an adenylyl cyclase-calcium (Ca^2+^)-RhoA pathway. The importance of this cholesterol–TAZ pathway remains to be demonstrated in human NASH [59].

In contrast, dietary PUFAs and fibers have beneficial effects on NAFLD. Saturated FAs increase fat accumulation in the liver, whereas the consumption of monounsaturated FAs (MUFAs) and PUFAs has protective effects against NAFLD [60,61,62]. PUFAs inhibit hepatic lipogenesis through repression of genes involved in lipid synthesis and glucose metabolism [63,64,65]. Epidemiological studies indicate that patients with NASH tend to have a higher intake of saturated FAs and cholesterol, whereas their PUFA consumption is lower compared to healthy controls [66].

A single oral dose of saturated FAs increases hepatic TG accumulation and impairs insulin sensitivity in human and mouse liver [67]. Compared to fat and carbohydrates, the role of dietary proteins in NAFLD pathogenesis is poorly studied and remains controversial. A recent study analyzed the effects of isocaloric diets rich in animal proteins or plant proteins for 6 weeks in patients with diabetes and NAFLD and found that both high-protein diets reduced liver fat [68]. However, high consumption of proteins and of red and/or processed meat has been reported in patients with NASH [69,70]. The Rotterdam study further showed that a high animal protein intake is associated with a higher prevalence of hepatic steatosis in overweight and elderly patients [71].

A role for micronutrients in NAFLD is less known, but the liver is important for their metabolism, which is often disturbed in liver diseases. Several micronutrients, such as zinc, copper, iron, selenium, magnesium, vitamins A, C, D, and E, and carotenoids have beneficial effects in NAFLD, mediated by their lipoprotective, antioxidant, antifibrotic, and immunomodulatory properties. However, the appropriate dosage appears to be important because an excess of iron and selenium may increase the severity of NAFLD [38,72]. Hepatic iron accumulation in reticulo-endothelial cells occurs in NAFLD and is associated with disease severity, but observations are often conflicting [73,74,75,76,77]. Similarly, some studies have reported that iron depletion by phlebotomy leads to histological improvements and liver enzyme normalization in NAFLD patients with hyperferritinemia, but results of several studies do not support the use of phlebotomy in these patients [78,79,80]. A classic intervention in patients with NAFLD is vitamin E supplementation because of increased oxidative stress. However, side effects, including hemorrhagic stroke and prostate cancer, limit the clinical use of vitamin E. Interventions with combinations of micronutrients are potentially interesting because experimental studies in mice have indicated beneficial effects on liver steatosis, body weight gain and hypertriglyceridemia, with findings suggesting that a cost-effective combinatorial micronutrient-based strategy could be tested in humans [81,82].

Probiotics are widely used to promote human health as described in a recent review [83], and prebiotics are nutrients used by host micro-organisms. They modulate the composition of the gut microbiota involved in NAFLD development [84,85]. Impacts of probiotic-based therapies in NAFLD patients are still debated. In one study, treatment of NAFLD patients with a combination of probiotics and prebiotics for 1 year led to faecal microbiome changes without modification of liver fat content and liver fibrosis markers [84]. In another study, NAFLD patients consuming a symbiotic food combining prebiotics and probiotics for 24 weeks presented improved serum steatosis associated parameters and oxidative stress biomarkers [86]. In a third study, treatment of Asian NAFLD patients with probiotics and prebiotics caused a decreased body weight and an amelioration of NAFLD parameters [85].

In animal models, pro- and prebiotic treatments also improved NAFLD development. Rat fed HFD containing the probiotic strain *Lactobacillus mali* APS for 12 weeks showed a decrease in body weight gain associated with reduced hepatic lipid accumulation and increased antioxidant response; these effects were associated with a decrease of some bacteria species [87]. Standard chow diet enriched with prebiotics (oligofructose or dextrin) induced changes in rat redox status and serum lipid biomarkers suggesting a positive impact of these prebiotics on NAFLD [88]. Cranberry extract could be another promising prebiotic in NAFLD prevention and/or treatment. Mice fed a high fat/high sucrose (HFHS) enriched with cranberry extract for 8 weeks improved all biomarkers of NAFLD [89]. Collectively, these results show that pre- and probiotics can ameliorate hepatic steatosis by modulating lipid metabolism and antioxidant activity.

The strong association between NAFLD and obesity further reinforces the role of dietary factors in the pathogenesis of NAFLD. Hence, in the absence of an approved drug treatment, as noted, a lifestyle intervention that targets weight loss is the primary therapy for the management of NAFLD. The effectiveness of weight reduction was recently confirmed in a meta-analysis of 22 randomized clinical trials that included more than 2500 patients with NAFLD. The results showed that weight loss intervention always improved their serum and histologic parameters, although no changes in liver fibrosis were observed [90].

In general, patients with NAFLD have a higher daily energy intake compared with healthy controls [91], and calorie restriction leading to weight loss has beneficial effects in these patients [92]. Both acute and chronic calorie-restricted diets with either a low-fat or low-carbohydrate content are associated with reduced liver fat in patients with obesity [93,94]. Hypocaloric diets are also effective in improving NAFLD in patients with diabetes and children with obesity [95,96]. In experimental models, calorie restriction extends lifespan and reduces the hepatic steatosis associated with obesity [92,97,98].

Gut microbiota

Some nutrients known to promote NAFLD, such as fructose, affect the gut microbiota and cause dysbiosis (an altered gut microbiota composition). These changes impair the permeability of the gut barrier and trigger the development of metabolic syndrome and its associated perturbations. In contrast, intake of dietary fiber, probiotics, and prebiotics associated with physical training is effective in improving NAFLD [99,100,101].

Of note, the gut microbiota is also required for the hepatic clock daily oscillations that regulate metabolic gene expression for optimal liver function [102,103,104]. In fact, the liver and gut are in close anatomical and functional relation via the portal vein. Diet, gut microbiota, and the liver constitute an axis that can promote either liver health or NAFLD progression. Analyses of the variations in gut microbiota composition in NAFLD have identified specific bacterial species that directly impact NAFLD progression [105]. In support of a beneficial effect, the administration of butyrate, a short chain FA produced by the gut microbiota, improves hepatic inflammation and fat accumulation in NAFLD mouse models [106]. However, despite many animal studies demonstrating a relationship between dysbiosis and NAFLD, only a limited number of cross-sectional human studies have investigated the role of the gut microbiota in NAFLD or NASH, with variable results [107]. Intestinal microbiota dysbiosis also has been associated with other liver diseases in animals and humans, including alcohol-related liver disease, cirrhosis, and HCC [107].

Food contaminants: the example of pesticides

Pesticides are considered as risk factors for human health. Various epidemiological studies reported a correlation between occupational exposure to these products and the development of pathologies, such as cancers, neurodegenerative diseases, and metabolic disorders [108,109,110,111]. Moreover, recent population studies (Nutrinetcohort in France) showed that the consumption of organic food correlated with a significant decreased risk of metabolic disease and cancer [112,113,114]. In addition, pesticides are bioactive compounds that exert an impact on various pathways underlying diseases [115]. Environmental contaminants are considered to play major roles in the etiology and progression of NAFLD. Endocrine-disrupting compounds were first described as inducing reproductive disorders but can be involved in the etiology of diabetes and other metabolic disorders independently of their hormone signaling effects [116,117,118,119,120,121,122]. In addition to contaminants such as polychlorinated biphenyls (PCBs), other compounds such as dioxins [123], bisphenol A [122], and pesticides are suspected to contribute to the increased prevalence of NAFLD.

Organochlorine insecticide levels in serum from adult participants of the National Health and Nutrition Examination Survey have been associated with increased odds ratios for alanine amino transferase (ALT), which may indicate NAFLD [124]. It is noteworthy that most epidemiological studies do not present information linking exposure to pesticide to NAFLD, but rather focus on diabetes and obesity [115,125]. In epidemiological studies, the serum concentrations of organochloride pesticides have been significantly associated with diabetes, especially in women [126], with a specific metabolic profile including mainly metabolites related to lipid metabolism [127], metabolic syndrome [128], and adiposity [129]. However, no correlation was found between serum levels of the organochloride dichlorodiphenyldichloroethylene (DDE) and the incidence of type 2 diabetes in a population of adults living in urban India [130]. Persistent but also non-persistent pesticides currently in use may also support the development of metabolic diseases [125,131,132,133,134]. Experimental studies have allowed identification of the type of pesticides that promote fatty liver disease or changes in global metabolic homeostasis in animal models [123], such as organophosphorus or neonicotinoid or pyrethroid insecticides [135,136,137,138,139,140,141,142,143,144,145,146], triazine herbicides [147,148], glyphosate herbicides [149,150,151,152,153], or other families [154,155] and fungicides [152,156,157,158,159,160,161] (Table 1).

Perinatal exposure to some pesticides has been correlated with metabolic changes [162], increased insulin levels in newborns [163], and overweight in childhood [164,165]. Experimental studies evaluating the consequences of pesticide exposure during the critical window of pre- and postnatal development (pregnancy and lactation) confirm that early life exposure to pesticides may have a role in metabolic disruption later in life (Table 1). A metabolic impact of perinatal exposure to pesticides in animal models has been reported for insecticides, such as chlorpyrifos [140,166] and imidacloprid [142]; herbicides, such as glyphosate [150] and 2–4 dichlorophenoxyacetic acid (2,4-D) [155]; and pesticide mixtures [167,168]. The metabolic consequences of early life exposure to organophosphate have been reviewed previously [61,74,169]. Early life exposure to fungicides has been tied to metabolic disturbances, reduced liver weight, and histopathological changes [170].

Pesticide exposure may lead to NAFLD through an impact on lipid metabolism by (i) modifying FA uptake and efflux [171] and FFA transport, (ii) increasing lipogenesis [172,173,174] (Table 1), (iii) altering oxidation pathways [175] (Table 1 and Table 2), and (iv) interacting with nuclear receptors involved in the control of metabolism [176,177,178]. Some pesticides may disrupt lipid metabolism [179,180].

Pesticides alter glucose metabolism through the activation of glucose uptake, glycogenolysis, gluconeogenesis (for insecticides, see [136]; for diazinon or monocrotophos-treated animals, see [181,182,183], or inhibition of the mitochondrial respiratory chain [184]. They also have exerted effects through modulation of carbohydrate response element binding protein (*Chrebp*) gene expression levels [185] and changes in the expression of peroxisome proliferator-activated receptor (PPAR)β/δ and genes involved in glucose metabolism (*FoxO1* and cAMP response element-binding protein, *Creb*) [186]. Pesticides induce insulin resistance by acting on insulin signaling pathways [187]. Liver inflammation was also reported upon exposure to organophosphorus [187,188], organochlorine [189], neonicotinoid [142,143], and pyrethroid insecticides [144,145], and to triazole or imidazole fungicides [190,191] (Table 1). 

**Table 1 biomolecules-10-01702-t001:** Metabolic impact of pesticide exposure in normal diet-fed animal models. Acceptable Daily Intake (ADI) values were from https://ephy.anses.fr/ and https://ec.europa.eu/food/plant/pesticides/eu-pesticides-database/. ADI, acceptable daily intake; ALT, alanine aminotransferase; AST, aspartate aminotransferase; BW, body weight; CAR, constitutive androstane receptor; GD, gestational day; HFD, high-fat diet; IL1-β, interleukin-1 beta LDH, lactate dehydrogenase; MDA, malonaldehyde; NOAEL, no observable adverse effect level; ND, normal diet; NF-ĸB, nuclear factor kappa B, PND, postnatal day; SOD, Super oxide dismutase; TG, triglyceride, TNF-α, tumor necrosis factor alpha; ADI, Acceptable Daily Intake.

Type of Pesticide	Chemical Family	Active Substance (ADI mg/kg BW/day)	Experimental Model	Metabolic Effects	Refs.
Insecticide	Organophosphorus	Diazinon (0.002)	ND-fed male rats Oral daily gavage with 15 mg/kg for 4 weeks	Increased serum ALT and AST activity, serum lipid content, peripheral inflammation biomarkers	[188]
		Malathion (0.03)	ND-fed male rats Oral daily gavage with 200 mg/kg for 28 days	Increased levels of lipid peroxidation biomarkers Decreased levels of antioxidant enzymes Chronic inflammation	[192]
			ND-fed male rats Oral daily gavage with 27 mg/kg for 30 days	Increased liver MDA levels Decreased liver glutathione levels, superoxide dismutase, and catalase activities Increased liver IL1-β, TNF-α, and NF-ĸB mRNA expression levels	[139]
Insecticide	Organophosphorus	Chlorpyrifos (0.001)	ND-fed pregnant rats Oral gavage with 1, 2.5, or 4 mg/kg BW/day from gestational day 7 to postnatal day 21	Increased adiposity and weight gain only in males with 2.5 mg/kg chlorpyrifos	[140]
			ND-fed female rats Oral daily gavage with 1 or 3.5 mg/kg BW/day during gestation and lactation	Decreased offspring BW gain at PND 60 Increased fasting glycemia Increased ALT activity Decreased plasma TG levels Change in microbiota composition	[166]
	Organochlorine	Endosulfan (0.006)	ND-fed male mice Oral daily gavage for 2 weeks with doses ranging from 0.5 to 3.5 mg/kg BW	Weight loss Changes in various liver metabolic pathways (energy, amino acid, and lipid metabolism) Gut microbiota alteration	[134]
	Neonicotinoid	Imidacloprid (0.06)	ND-fed female rats Oral daily gavage with 9 or 45 mg/kg BW for 4 weeks	Biomarkers of hepatotoxicity	[141]
		Imidacloprid (0.06)	ND-fed pregnant rats Oral daily gavage with 1/45th or 1/22th LD_50_ from mating to gestation and lactation	Hepatic necrosis and inflammation in the non-exposed second generation	[142]
		Thiamethoxam (0.026)	ND-fed male rabbits Oral daily gavage with 250 mg/kg BW for 90 days	Hepatic oxidative stress and inflammation	[143]
Insecticide	Pyrethroid	Deltamethrin (0.01)	ND-fed male rats Oral daily gavage with 15 mg/BW for 30 days	Liver inflammation (increased levels of serum and liver lipid peroxidation biomarkers) Changes in antioxidant enzyme activity Increased expression of inflammatory *cox*2 gene	[144]
		Alpha cypermethrin (0.00125)	Female rats Pesticide-enriched diet at 0.02 mg/kg BW/day One month before and during gestation	Increased glucose, cholesterol, TG levels and AST and ALT activity in mothers and newborns Increased maternal BW and liver weight in mothers Change in pup liver weight Increased liver inflammation and lipid content in both dams and newborns Increased circulating and liver levels of oxidative stress biomarkers and decreased antioxidant status in both dams and newborns	[145]
		Alpha cypermethrin (0.00125)	Pregnant female ND-fed rats Pesticide enriched diet (1.5 mg/kg) During gestation and lactation and for 5 months after weaning	Reduction of body weight, food and energy intake in offspring Increase in plasma glucose, urea cholesterol and creatinine levels in both male and female rats Increase in liver oxidative stress	[146]
Insecticide	Pyrethroid	Lambda cyhalothrin (0.0025)	ND-fed male rats Oral daily gavage with 6.2 or 31.1 mg/kg BW for 7, 30, 45, or 60 days	Increased levels of hepatic stress biomarkers (MDA, AST, ALT) and antioxidant enzyme activities Increased liver TNF-alpha and interleukin gene expression	[137]
			ND-fed male rats Gastric intubation with 1, 2, 4, and 8 mg/kg BW for 6 consecutive days	Changes in the expression of genes coding for xenobiotic metabolism enzymes and for liver oxidation	[138]
Herbicide	Triazine	Atrazine (0.02)	HFD- and ND-fed male rats Oral exposure in drinking water 30 to 300 µg/kg BW/day for 5 months	BW gain in HFD- and ND-fed rats Increased insulin levels in HFD-fed rats Insulin resistance in ND- and HFD-fed rats	[148]
	Glycine derivate	Glyphosate (0.5)	Pregnant ND-fed mice Oral exposure to 0.5% glyphosate solution in drinking water during gestation and lactation ND-fed male rats Oral exposure to 0.1 ppb Roundup formulation (eq to 0.05 µg/L glyphosate) in drinking water for 2 years	Decreased body weight in offspring PND7 and 21 Increased lipid levels in offspring in liver and plasma Liver proteome disruption reflecting steatosis and necrosis	[150,151]
Herbicide	Glycine derivate	Glyphosate (0.5)	ND-fed male rats Oral daily gavage with 5, 50, or 500 mg/kg BW for 35 days	Increased levels of serum and liver MDA Decreased liver SOD activity Increased liver catalase activity Increased liver mRNA expression of inflammatory genes	[153]
	Dinitroaniline	Pendimethalin (0.125)	ND-fed male rats Oral daily gavage with 62.5, 125, or 250 mg/kg BW/day for 14 days	Increased biomarkers of liver lipid peroxidation and protein carbonylation Decreased antioxidant enzyme activities Liver hyperplasia and swelling, occurrence of pyknotic nuclei, activated Kupffer cells and leukocyte infiltrations, large cytoplasmic vacuolization and dilatation in blood sinusoid	[154]
		2.4 Dichlorophenoxyacetic acid (2,4-D) (0.02)	ND-fed pregnant rats Co-exposure through drinking water at 126 mg/kg BW from GD 14 to PND 14 Observation at PND 14	Decreased pup BW Decreased mother BW and liver weight, but also in food and water intake Increased plasma ALT, LDH, and AST activity in both pups and dams Increased MDA levels and antioxidant enzyme activities in dams and pups Liver inflammation in both dams and pups	[155]
Fungicide	Triazole	Penconazole (0.03)	ND-fed male mice Oral exposure to 30 mg/L in drinking water for 4 weeks	Increased levels of serum ALT, AST and hepatic TGs, total cholesterol; lipid droplet accumulation and inflammation	[190]
		Cyproconazole (0.02)	ND-fed WT and humanized CAR male mice Pesticide-enriched diet with 50 or 500 ppm	Induction of CAR-dependent gene expression Increased liver weight Hepatic lipid accumulation	[157]
	Triazole	Cyproconazole (0.02) Epoxiconazole (0.008) Prochloraz (0.01)	ND-fed male rats Pesticide-enriched diet: individual compounds or mixture (NOAEL/100; NOAEL; NOAELx10)	Changes in the expression of hepatic genes involved fatty acid and phospholipid metabolism, cytochrome P450, transporters (Abcb1a, Abcc3) In most cases, treatment with mixtures caused stronger effects as compared to the individual substances	[158]
	Imidazole	Imazalil (0.025)	ND-fed male mice Pesticide-enriched diet with 25, 50, or 100 mg/kg BW daily for 4 weeks	No significant changes in hepatic TG Dysbiosis and colonic inflammation	[160]
	Benzimidazole	Carbendazim (0.02)	ND fed male mice 0.1, 0.5, or 5 mg/kg BW per day in drinking water for 14 weeks	Increased intestinal fat absorption Reduced liver lipid synthesis Increased fat lipid storage Intestinal dysbiosis Inflammatory response in various tissues	[191]
Fungicide	Sulfamide	Tolylfluanid (0.1)	ND-fed male mice Pesticide-enriched diet with 100 ppm for 12 weeks	Increased BW and adiposity, glucose intolerance, and insulin resistance, impaired circulating levels of leptin and adiponectin, disturbed fat oxidation and lipolysis in adipocytes No change in liver homeostasis	[161]
Mixture Insecticide and fungicide	Triazine Phenoxybutyric Carbamate Quinone Dithiocarbamate Carbamate	Cyromazine (0.06), MCPB (0.01), Pirimicarb (0.035), Quinoclamine (0.002), Thiram (0.01), Ziram (0.006)	ND-fed pregnant female rats Oral gavage with total dose of 28, 104, or 210 mg/kg/day of the mixture from GD 7 to GD 17 Analysis at 4 months of age	Decreased birth weight in exposed male offspring Increased leptin levels in female offspring Variations Change in the expression of 3 genes in fat tissues in a sexually dimorphic manner	[167]
Mixture of insecticide, herbicide, and fungicide	Carboxamide Phthalimide Organophosphorus Neonicotinoid Dithiocarbamate Carbamate	Boscalid (0.04) Captan (0.1) Chlorpyrifos (0.001) Thiachloprid Thiophanate (0.08) Ziram (0.006)	ND-fed male and female mice Pesticide mixture-enriched diet, exposure to the ADI of each pesticide for 52 weeks	Overweight and glucose intolerance and steatosis in males Liver oxidative stress and alteration in gut microbiota in females	[193]
Mixture of insecticide, herbicide, and fungicide	Chloroacetanilide Nitrile Carbamate Quaternary ammonium Organophosphorus Morpholine Glycine derivative Neonicotinoid	Acetochlor (0.0036) Bromoxynil (0.01) Carbofuran (0.00015) Chlormequat (0.04) Ethephon (0.03) Fenpropimorph (0.003) Glyphosate (0.5) Imidacloprid (0.06)	ND-fed pregnant rats Pesticide-enriched diet; exposure at a nominal dose corresponding to the same proportion as their respective environmental exposure value (from French use) from GD 4 to GD 21	Decreased liver weight in dams Increased liver weight in male offspring Decreased liver lipid content in male offspring Changes in liver metabolome (lipid metabolism biomarkers) and serum metabolome in fetuses	[168]

Of interest, pesticide exposure and a high-fat diet (HFD) (Table 2) [136] interact to modify their effect on metabolic homeostasis [194]. In fact, various chemical families of insecticides seem to be involved, i.e., organochlorine, pyrethroid, neonicotinoid, and organophosphorus compounds (Table 2 [148,184,185,195,196,197,198,199,200,201,202,203,204,205,206,207,208,209,210]). Moreover, HFD components may act as a vehicle for pesticides, allowing them to reach target organs more easily, as proposed in [136,211]. This finding suggests that a different bioavailability of pesticides in mice fed with an HFD versus normal chow diet may explain, at least in part, dissimilar outcomes with pesticide exposure in different studies.

The fact that some pesticides have antimicrobial activity implies that they affect the composition and metabolic functions of the gut microbiota and may be metabolized by the microbiota [212,213]. For example, the organochloride dichloro-diphenyl-trichloroethane (DDT) is converted to DDD by the rat and human microbiota (*Eubacterium limosum*) [214]. When administered by oral gavage in mice, DEE, a metabolite of DDT, impacts bile acid metabolism and reduces the relative abundance of Bacteroidetes, Verrucomicrobia, and Actinobacteria, whereas that of Firmicutes and Proteobacteria is increased. Furthermore, genes involved in bile acid synthesis are upregulated in the liver [215]. Many studies mainly using rats have analyzed the impact of other insecticides on the microbiota, and important changes in its composition have been reported (reviewed in [213]). Relatively little is known about how glyphosate, the most widely used herbicide worldwide, affects NAFL in terms of gut microbiota, but glyphosate has a profound effect on both gut microbiota composition [216] and metabolism [217] in rodents. Roundup, whose active compound is glyphosate, is suspected to lead to greater alterations in the gut microbiota composition than glyphosate alone, alterations that are similar to those reported in NAFLD, obesity, and systemic inflammation [213,218,219]. Finally, fungicides also cause an important dysbiosis and, one of them, carbendazim, has been associated with hepatic oxidative stress [220]. Overall, pesticides in general are associated with important dysbiosis but the link among pesticides, dysbiosis, and hepatic oxidative stress in NAFLD is not well established, and further studies are warranted.

**Table 2 biomolecules-10-01702-t002:** Metabolic impact of pesticide exposure in high-fat diet-fed animal models. Acceptable Daily Intake (AD)I values were from https://ephy.anses.fr/ and https://ec.europa.eu/food/plant/pesticides/eu-pesticides-database/. ADI value is not reported for some compounds due to insufficient data (organochlorine compounds, prinomectin, moxidectin, GW4064, PCB). ALT, alanine aminotransferase; BW, body weight; FFA, free fatty acid; CPT, carnitine palmitoyl-transferase; FXR, farnesoid X receptor; GD, gestational day; GIP, glucose-dependent insulinotropic peptide; GLP-1, glucagon-like peptide-1; HFD, high-fat diet; HOMA-IR, homeostasis model assessment of insulin resistance. NEFA, non-esterified fatty acid; NOAEL, no observable adverse effect level; ND, normal diet; PCB polychlorinated biphenyl; PP, pancreatic polypeptide; PUFA, polyunsaturated fatty acid; PND, postnatal day; SFA, saturated fatty acid; TG, triglyceride; UCP, uncoupling protein; VLDL, Very low density lipoprotein; WT, wild type; ADI, Acceptable Daily Intake.

Type of Pesticide	Chemical Family	Active Substance (ADI mg/kg BW/day)	Experimental Model	Effects of Pesticide Exposure on Diet-Induced Metabolic Disorders	Refs.
Insecticide	Organochlorine	DDE	ND- and HFD-fed male rats Pesticide-enriched diet for 4 weeks	Increase serum ALT and AST levels in ND Decreased serum TG levels and liver lipid content in HFD fed animals Increased CPT activity oxidation and lipid peroxidation in ND- and HFD-fed animals Hepatic oxidative stress in ND- and HFD-fed animals Increased antioxidant enzyme activities more in ND- than in HFD-fed animals Increased *UCP2* mRNA levels in ND- and HFD-fed animals	[203]
			ND- and HFD-fed male rats exposed to 100 µkg/BW/day DDE in drinking water for 12 weeks	Increased accumulation in fatty acid (SFA and PUFA) content in both ND- and HFD-fed animals	[204]
Insecticide	Organochlorine	DDE	Oral gavage of ND-fed male mice for 5 days One-week resting period Then oral gavage each week of ND- or HFD-fed males for 13 weeks	After 4 and 8 weeks on the HFD diet, increased fasting hyperglycemia and liver steatosis At week 13, HFD-induced decrease in fasting hyperinsulinemia, HOMA-IR values, and hepatic steatosis	[205]
		DDT	Daily oral gavage of pregnant female mice from GD 11 to PND 5 at 1.7 mg/kg BW ND- and HFD-fed adult female offspring for 12 weeks	Reduced body temperature and energy expenditure and increased body fat in offspring of ND mothers Compared to ND, further HFD-induced reduction of body temperature HFD-induced changes in brown adipose tissue gene expression	[206]
		Chlordane	ND- and HFD-fed male mice (6 weeks) Oral gavage daily at 1.45 mg/kg BW from week 4 to week 6	No change in BW in ND- and HFD-fed animals Increase in HFD induced liver metabolic perturbations (TCA cycle, tryptophan catabolism, nucleotide, lipid and choline, and amino acid metabolism)	[207]
Insecticide	Pyrethroid	Permethrin (0.050)	ND- or HFD-fed male mice Pesticide-enriched diet with 50, 500, or 5000 µg/kg BW/day for 12 weeks	Potentiation of HFD-induced BW gain, total adipose tissue weight, insulin resistance	[208]
		Cypermethrin (0.050)	HFD-fed male mice simultaneously exposed to 50 µg/kg BW/day cypermethrin in drinking water for 20 weeks	No change in BW Increased levels of serum FFA, hepatic lipid, TG Increased liver *de novo* FFA and TG synthesis Increased uptake of FFA from blood	[185]
	Organophosphorus	Chlorpyrifos (0.001)	ND- and HFD-fed male mice Co-exposure by daily gavage with 5 mg/kg for 12 weeks	Increased BW in both ND- and HFD-fed mice Impaired glucose metabolism and insulin resistance in both HFD- and ND-fed mice	[209]
			ND- and HFD-fed male rats Co-exposure by daily oral gavage to 0.3 and 3 mg/kg BW for 9 weeks	Increased weight gain in ND- but not in HFD-fed rats Obliteration of the increase in plasma TG caused by HFD Decreased plasma insulin, C-peptide, and amylin concentrations in both the ND- and HFD-fed rats Decreased plasma levels of leptin GIP and GLP-1 and PP induced in HFD-fed rats Differential alteration of gut microbiota composition according to the diet	[210]
Insecticide	Organophosphorus	Chlorpyrifos (0.001)	ND- and HFD-fed male mice for 4 weeks Oral gavage with 2 m/kg BW/day chlorpyrifos during the last 10 days of the experiment	No significant body weight increase in HFD-fed animals No significant increase in plasma TG in HFD-fed animals Decreased expression of genes involved in liver lipogenesis in ND- and HFD-fed animals	[195]
			HFD- and ND-fed male mice for 4 weeks Chlorpyrifos (2 mg/kg) single oral gavage at the end of the experiment	Increased TG serum levels in ND fed animals Hypoglycemia in HFD-fed animals.	[196]
		Parathion (0.0006)	Subcutaneous injection of PDN 1–4 in rats at 0.1 and 0.2 mg/kg/day once daily A 6 week HFD feeding of 15 week-old male and female rats	Increased serum cholesterol levels in fasted HFD-fed males Decreased serum NEFA levels in fasted ND-fed females Enhancement of HFD-induced BW gain in females exposed to 0.1 mg/kg/day parathion Decreased BW gain in HFD fed females exposed to 0.2 mg/kg BW parathion	[197]
Insecticide	Organophosphorus	Acephate (0.03)	Daily gavage of female rats with 2.5 mg/kg BW/day from pregnancy day 7 until lactation day 21 Overfeeding of pups by decreasing the number of pups per nest Analysis at PND 90	Decreased BW in both animal groups at PND 1 Decreased BW in overfed animals at PND 90 Increased peritoneal and mesenteric fat mass only in normal fed animals at PND 90 Decreased energy intake in overfed animals at PND90 Increased fasting glucose in normal- and overfed animals Increased fasting insulin in normal-fed animals Increased glucose intolerance and insulin resistance in normal-fed animals Reversion of many of the programmed postnatal overfeeding induced metabolic disturbances at PND 90	[198]
	Neonicotinoid	Imidacloprid (0.06)	ND- and HFD-fed male mice Pesticide-enriched diet Exposure to dose ranging from TDI to NOAEL for 12 weeks	Potentiation of HFD-induced weight gain, insulin resistance, and adipocyte size and impaired glucose metabolism	[199]
Insecticide antiparasitic	Avermectin	Abamectin (0.0025) Doramectin, (0.0005) Ivermectin (0.010), Eprinomectin Moxidectin GW4064	HFD-fed male mice simultaneous exposed to 1.3 mg/kg B avermectin analogs by intraperitoneal injection for 14 days	Suppression of HFD-induced metabolic disturbances in mice by most compounds, excepting eprinomectin and moxidectin	[200]
		Ivermectin (0.010)	HFD fed WT and FXR null mice simultaneous exposed to ivermectin by intraperitoneal injection at 1.3 mg/kg BW/day for 14 days	Decreased serum levels of glucose and cholesterol, high-density lipoprotein and low-density lipoprotein and VLDL cholesterol levels, in wild-type mice Ivermectin effects suppressed in FXR null mice	[201]
Fungicide	Strobilurin	Azoxystrobin (0.2)	HFD-fed male mice for 13 weeks Oral gavage at 25 mg/kg BW/day at week 8 for 5 weeks	Reduction in BW Improvement of glucose tolerance Decrease liver TG accumulation Decrease liver lipogenesis	[184]
Herbicide	Triazine	Atrazine (0.02)	HFD-fed male mice simultaneously exposed to atrazine in drinking water at 100 µg/kg BW/day for 20 weeks	No change in BW Increased levels of serum FFA, hepatic lipids, and TG Increased liver *de novo* FFA and TG synthesis pathways Increased uptake of FFA from blood	[221]
			HFD- and ND-fed male rats simultaneously exposure to atrazine in drinking water with 30 to 300 µg/kg BW/day for 5 months	Body weight gain in HFD- and ND-fed rats Increased insulin levels in HFD-fed rats Insulin resistance in ND- and HFD-fed rats	[148]
Mixture	POP	Organochlorine pesticide and PCB mixture	Leptin-deficient ob/ob ND fed male mice Oral gavage twice weekly for 7 weeks at environmentally relevant levels	Increased steatosis in ob/ob mice Increased hepatic triglyceride content in ob/ob mice	[202]

Other environmental factors can affect NAFLD development and progression. Although the effect of active smoking on NAFLD development remained controversial [222] a recent systematic review and meta-analysis from 20 observational studies revealed a slight but significant association between active and passive smoking and NAFLD [223]. The link between a modest alcohol consumption with NAFLD development is described in a very recent review [224].

Aim of the review

There is currently no effective pharmacological therapy for the treatment of NAFLD. Weight loss, achieved through lifestyle changes including diet modifications and exercise, remains the most effective strategy for NAFLD management. However, these lifestyle interventions are difficult to maintain for most patients. Vitamin E and the anti-diabetic agent Pioglitazone are the two classic therapies currently used to treat NAFLD patients but they can both induce adverse effects and their benefit on liver fibrosis is either absent or still not clear. Most of the other molecules that are under investigation target the metabolic comorbidities of NAFLD such as obesity, insulin resistance, and dyslipidemia [225]. A better understanding of the development and progression of this complex pathology will allow the development of novel therapeutic strategies for the treatment of NAFLD.

As is well known, alterations in lipid metabolism are key to the development of NAFLD and its progression. They can affect different reactive oxygen species (ROS) generators, such as mitochondria, endoplasmic reticulum (ER), and nicotinamide adenine dinucleotide phosphate (NADPH) oxidase (NOX) [226]. Although many aspects of the contributions of these disturbances to NAFLD remain unexplored, much progress has been accomplished in recent years. Knowledge is accumulating on how increased ROS generation triggers changes in insulin sensitivity and the activity of key enzymes of lipid metabolism, innate immune system signaling, and the level of inflammatory responses. Our review addresses the mechanisms and consequences of excessive ROS production in the liver that drives NAFLD progression, and the roles of nutrients and food contaminants in these processes (Figure 1).

Our discussion on the roles of oxidative stress on NAFLD development covers knowledge gained from both animal models, mainly rodent models, and humans. It is important to keep in mind, while reading the review, that rodents and humans NAFLD are not equivalent, although the FDA encourages the use of appropriate animal models for developing new treatments against pre-cirrhotic NASH [227]. The current animal models mimicking the progression of the human NAFLD phenotype rely on genetic alterations, chemically induced liver damage or diets deficient in essential nutrients. Therefore, they do not fully match, mechanistically and biochemically, the lifestyle-associated NAFLD in humans [228]. There are several models used currently, each presenting advantages and limitations [229,230]). A useful animal model, from a clinician’s view point, should reflect the important metabolic deteriorations seen in human NAFLD patients including hyperglycemia and insulin resistance, hyperlipidemia, and obesity with liver fat accumulation. Fatty liver should show progression from simple steatosis to the inflamed liver stage with ballooning and fibrosis monitored by histopathological examination and quantified by recognized scoring systems. The progression should be relatively fast allowing the evaluation of candidate drugs and nutritional interventions. Despite relentless efforts and the advances made in recent years, the ideal rodent model comprising all desirable characteristics is not yet available. Therefore, studies in different models are necessary to perform all the analyses required for a better understanding of human NAFLD progression.

## 2. Methods

We performed a PubMed search of the literature. Studies that were not published in English or published in the grey literature were excluded. We tried to focus on articles published during the last 10 years. We established a list of keys words (below) and combined them according to each author’s dedicated chapter in order to get a representative overview of the literature. Each author reviewed the chapter of the other co-authors. Furthermore the corresponding authors, worked on the homogeneity and uniformity of the different chapters. The following lists of keywords were used:(i)“Oxidative stress” OR “oxidative phosphorylation” OR “oxidation-reduction potential” OR “reactive oxygen species” OR “ROS” OR “reactive nitrogen species “ OR “RNS” OR “redox stress” OR “peroxidation” OR “free radicals” OR “lipotoxicity” OR mitochondria OR endoplasmic reticulum OR glutathione OR antioxidant pathways OR Nrf2 OR SIRT OR SOD OR GPX OR Catalase(ii)“metabolic hepatitis” OR “metabolic liver” OR “Non Alcoholic Fatty Liver Disease” OR “Nonalcoholic Fatty Liver Disease” OR “Fatty Liver” OR “Nonalcoholic Fatty Livers” OR “Nonalcoholic Liver” OR “Nonalcoholic Fatty” OR “Nonalcoholic Fatty Liver” OR “Nonalcoholic Steatohepatitis” OR “Nonalcoholic Steatohepatitides” OR “Steatohepatitides” OR “Nonalcoholic Steatohepatitis” OR “cirrhosis” OR “carcinogenesis” OR “hepatocarcinoma” OR “hepatocellular carcinoma” OR “carcinoma” OR “hepatoma” OR malignant hepatoma”(iii)“mice” OR “mouse” OR “mus musculus“ OR “clinical studies” OR “therapeutic”(iv)“pesticides” OR “herbicides” OR “insecticides” OR “fungicides” OR “plant protection product”(v)“oxidant” OR “antioxidant” OR “diet” OR “cholesterol” OR “fructose” OR “nutrients” OR “micronutrients”, “macronutrients” OR “fat” OR “western diet” OR “sugars” OR “fibers” OR “vitamins” OR “fatty acid”.(vi)“Probiotic” OR “microbiota” OR “dysbiosis”.

## 3. Role of Oxidative Stress in NAFLD Pathogenesis

Oxidative stress is defined as an imbalance between the cellular levels of antioxidants and that of pro-oxidants, including ROS and reactive nitrogen species (Figure 2), which causes cellular damage and, in most cases, cell death. In healthy physiological conditions, cells maintain a basal level of ROS to promote the balanced redox signaling required for various processes such as cell metabolism, cell differentiation and survival, immune defense, and modulation of transcription factor activity and epigenetic state [226]. Upon oxidative stress, the antioxidant enzyme superoxide dismutase (SOD) generates hydrogen peroxide (H_2_O_2_) from the superoxide radical (O_2_^.−^), which is then processed to oxygen (O_2_) and water (H_2_O) through glutathione peroxidase (Gpx) or catalase enzyme activities (Figure 2) [231]. Within cells, ROS are mainly produced in the mitochondria, the peroxisomes, and the ER, but cytoplasmic production of ROS also occurs. High levels of ROS alter these organelles, further enhancing oxidative stress and creating a vicious circle.

### 3.1. Evidence for a Role of Oxidative Stress in NAFLD

Oxidative stress promotes activation of enzymatic or non-enzymatic-mediated antioxidant mechanisms that counteract ROS production. Both clinical and experimental studies show that these antioxidant pathways are modulated during NAFLD progression. In fact, activity of the antioxidant enzymes SOD and Gpx increases in patients with NAFLD (Figure 2, Figure 3, Figure 4 and Figure 5) [232]. In vitro, hepatic stellate cells deficient for the glutathione peroxidase 7 (*Gpx7*) isoform present increased expression of profibrotic and pro-inflammatory genes in response to FFA exposure. Consistent with these results, overexpression of *GPX7* in these cells decreases ROS generation and expression of profibrotic and pro-inflammatory genes. In vivo*, Gpx7* deficiency exacerbates choline-deficient, L-amino–defined, high-fat diet-induced NASH fibrosis [233]. The expression of glutaminase 1 (*GLS1*) is increased in both NASH preclinical mouse models and clinical NASH liver biopsies. GLS1 promotes glutamine fueling of anaplerotic mitochondrial metabolism resulting in increased ROS production. In methionine choline-deficient (MCD) diet-fed mice, *GLS1* inhibition decreases hepatic TG accumulation by restoring VLDL TG export and diminishes oxidative stress by lowering ROS production. GLS1 deficiency in this model is also associated with decreased lipid peroxidation [234]. Paraoxonase-1 is a liver antioxidant enzyme that hydrolyses peroxide and lactones associated with lipoproteins. In a cohort of 81 patients with NAFLD, the serum Paraoxonase-1 concentration was decreased, which could reflect higher oxidative stress in these patients [235].

The peroxisomal antioxidant enzyme catalase plays a key role in protecting cells from oxidative damage by reducing H_2_O_2_ concentration (Figure 3). In HFD-fed mice deficient for catalase, lipid accumulation and oxidative stress are exacerbated [236]. These mice develop an imbalanced redox status in peroxisomes because of increased H_2_O_2_ levels, which in turn induces ER stress in the liver. Catalase inhibition in HepG2 cells increases ROS production by peroxisomes, causing ER stress and FA accumulation [237]. In line with this observation, human liver cells with impaired peroxisome biogenesis show decreased ER stress, oxidative stress, and apoptosis [238].

Expression of genes encoding antioxidant enzymes and regulators of the glutathione pathway (glutamate-cysteine ligase catalytic subunit, glutamate-cysteine ligase modifier subunit, GPX2) is regulated by the transcription factor erythroid2-like 2 (Nrf2) [239]. Upon oxidative stress, Nrf2, which is anchored in the cytoplasm via binding to Keap1, dissociates from it and translocates into the nucleus, where it interacts with specific DNA sequences called antioxidant response elements in the promoter of its target antioxidant enzyme genes [240]. *Nrf2* expression is increased in the first stage of NAFLD in preclinical models [241], and pharmacological activation of Nrf2 in mice fed a high-fat and high fructose diet decreases NASH parameters (insulin resistance, weight gain, TG, ALT) via the transcriptional regulation of genes involved in inflammation, apoptosis, fibrosis, ER stress, and oxidative stress [239]. Consistently, *Nrf2* deficiency in mice fed an MCD or an HFD promotes progression of steatosis to NASH by increasing oxidative stress, inflammation, and hepatic FA accumulation [242,243]. In addition, *Nrf2*-deficient mice fed an HFD develop a more severe NASH phenotype than their WT counterparts [241]. In in vivo and in vitro models of NAFLD, the reduction of *Nrf2* expression by microRNA is associated with a decreased expression of its target genes, heme oxygenase, *Sod2*, and NAD(P)H dehydrogenase quinone 1 and an increase in ROS production [244]. Nrf2 also directly affects lipid metabolism by activation of genes involved in FA oxidation (acyl-CoA Oxidase 2, carnitine palmitoyltransferase 1), TG export (apolipoprotein B), and the lipogenic transcription factor sterol regulatory element binding transcription factor 1 (*Srebp-1*) [239]. Nfr2 deficiency in HFD-fed mice diminishes phosphorylation of acetyl-CoA carboxylate (ACC), a rate-limiting enzyme of hepatic FA synthesis, and thus increases its activity [241]. Collectively, these data indicate that alterations in antioxidant pathways are associated with NAFLD, suggesting a role of oxidative stress in disease progression.

### 3.2. Mechanisms of Oxidative Stress in NAFLD

#### 3.2.1. Mitochondria-Mediated Oxidative Stress

Mitochondria are intracellular sites of oxygen (O_2_) consumption. In case of energy imbalance, such as lipid accumulation, mitochondria ROS production increases considerably because of mitochondrial respiratory chain alteration and a reduction in electron capture through various mechanisms discussed below (Figure 4). Generation of mitochondrial ROS results in impairment of the mitochondrial membrane potential generated by proton pumps and activation of the JNK (c-Jun N-terminal kinase) and AMPK (5’ AMP-activated protein kinase) pathways. All of these mechanisms enhance oxidative stress and lipid accumulation and promote inflammation, thereby contributing to the development of obesity and metabolic diseases, including NAFLD [245].

Mitochondrial ROS production results from dysfunction in complexes I and II of the mitochondrial respiratory chain. Various mechanisms leading to this dysfunction and ROS production have been described in NAFLD. In ob/ob mice, the quantity of mitochondrial respiratory chain complex I is decreased. In this model, mitochondrial oxidative stress causes mitochondrial DNA (mtDNA) damage, resulting in the reduction of mtDNA-encoded subunits of respiratory chain complex I. mtDNA is in the mitochondrial matrix near the mitochondrial respiratory chain where most ROS are generated. Thus, it is particularly prone to oxidative damage compared with nuclear DNA, which is more protected inside the cell nucleus. Furthermore, mtDNA lacks protective histones, which together with the absence of an mtDNA repair system, makes it more susceptible to damage [246]. Mitochondrial complex I dysfunction can also arise from mitochondrial protein 3-tyrosine nitration by peroxynitrite anion. Peroxynitrite originates from a combination of nitric oxide (NO^.^) with superoxide (O_2_**^.^**^−^), which are both elevated in ob/ob mice and in patients with NAFLD. In vitro, incubation of mitochondrial proteins from wild-type (WT) mice with peroxynitrite induces their 3-tyrosine nitration, which results in decreased complex I activity and mitochondrial oxygen consumption. This decrease is amplified by reduced expression of prohibitin, a protein that protects mitochondrial complexes against degradation [246]. Complex I or II activity is also regulated by the mitochondrial histone deacetylase sirtuin (SIRT) 3. Mice fed an HFD have decreased SIRT3 expression, which results in a higher level of membrane transport chain complex I acetylation, causing mitochondrial dysfunction. MCD-fed mice deficient in SIRT3 develop more severe liver lesions, inflammation, and fibrosis compared with WT mice [247]. The proper functioning of the mitochondrial respiratory chain requires ATP, and in turn, impaired ATP production leads to transport chain complex dysfunction. In NAFLD, oxidative phosphorylation (OXPHOS) and the TCA cycle are disturbed, which causes impaired ATP production leading to an alteration in mitochondrial respiration and ROS production. In patients with NASH, antioxidant capacity and ATP production decreases, which enhances hepatocellular damage and insulin resistance, promoting NAFLD progression [248].

Mitochondria are dynamic organelles, and fission and fusion are crucial for maintaining their function under environmental stress. Both processes are regulated by specific proteins, Fis-1 and Drp-1 for fission and Mfn-2 for fusion, which show decreased expression in WD-fed mice, suggesting that mitochondrial dynamism is affected in NAFLD [249]. ROS accumulation in mitochondria leads to modification of several mechanisms associated with oxidative stress and NAFLD development. Phosphorylated JNK under lipotoxicity or ER stress conditions phosphorylates SH3-domain binding protein 5 at the outer mitochondrial membrane, leading to the inactivation of mitochondrial c-Src, which regulates the phosphorylation of respiratory chain components. This Src inactivation alters electron transport, which promotes increased ROS release [250]. Phosphorylated JNK is also associated with insulin resistance, which further suggests its involvement in NAFLD development [251]. Oxidative stress induced in HFD-fed mice causes liver damage through mitochondrial membrane potential alteration [245]. ROS accumulation in mouse models of NAFLD affects mitochondrial depolarization potential through the formation of aldehydes (malondialdehyde [MDA] and 4-hydroxynonenal [4-HNE]) by lipid peroxidation or an increase in mitochondrial sensitivity to Ca^2+^.

Mitochondrial depolarization is a dysfunction that appears early in NAFLD development and contributes to mitochondrial homeostasis deregulation. In fact, WD-fed mice show mitochondrial depolarization at an early stage of NASH, with an increased protein level of PINK1 (PTEN-induced kinase 1), a mediator of mitochondrial autophagy (mitophagy), which is associated with mitochondrial dysfunction. In these situations of mitochondrial depolarization, the mitophagic burden increases, mitochondrial biogenesis declines, and mitochondrial depletion occurs after 2 to 6 months. These alterations are thought to be important in promoting steatosis, inflammation, and progression to fibrosis [249].

The ROS H_2_O_2_, produced by mitochondria, activates AMPK through modification of the ATP/ADP ratio [252], which further regulates antioxidant enzyme gene expression through *Nrf2* activation [253]. The AMPK pathway may be activated by an antioxidant component Peroxiredoxin 5 (Prx5). Indeed, in HepG2 cells exposed to FFAs, mitochondrial Prx5 activates the AMPK pathway to regulate the activity of lipogenic enzymes (ACC, SREBP-1, FA synthase) [254]. AMPK stimulates glucose and FA oxidation by induction of *CPT-*1 and *acyl-CoA dehydrogenase* expression via PPAR gamma coactivator 1-alpha and PPARα [253]. Thus, although the AMPK pathway represents a protective response against oxidative stress and lipid accumulation, prolonged oxidative stress alters this pathway, leading to lipid accumulation and further enhanced oxidative stress. A recent study showed that the AMP kinase pathway is inhibited in NASH, leading to the activation of caspase 6-associated cell death [255]. Similarly, other antioxidant responses may have deleterious effects. Early upregulation of uncoupling protein-2 (*UCP-2*) during steatosis protects hepatocytes against ROS production through ATP production, whereas a further increase in *UCP-2* during NASH leads to chronic ATP depletion [256].

Overall, in NAFLD, mitochondria are an important source of ROS because of alterations in complex I and II activity that initially lead to the activation of antioxidant mechanisms. However, during prolonged oxidative stress, these mechanisms participate in oxidative damage and enhance mitochondrial oxidative stress, thus contributing to the development of NAFLD (Figure 4).

#### 3.2.2. ER-Mediated Oxidative Stress

The ER is an organelle connected to the nuclear membrane in eukaryotic cells and is abundant in hepatocytes because of their high metabolic activity. It has multiple functions including protein synthesis, folding, modification, and trafficking, and synthesis of lipids and steroid hormones. Alteration of ER homeostasis, i.e., ER stress, can induce oxidative stress (Figure 5). The ER stress response is defined by activation of the unfolded protein response (UPR) pathway. Prolonged ER stress increases the PKR-like ER protein kinase (PERK) and activated transcription factor (ATF) 6 pathways that both lead to ROS generation through the proapoptotic C/EBP homologous protein (CHOP), a specific protein of the ER stress response (Figure 5). CHOP is involved in oxidative stress induction in mouse models of type 2 diabetes and MCD-induced steatohepatitis [226,257]. Recently, ER stress was shown to be involved in lipogenesis and in the transition of NAFLD to NASH through caspase 2 induction [258]. ER also contributes to the regulation of calcium (Ca^2+^) homeostasis and particularly Ca^2+^ storage in the ER lumen. In animal models of diabetes and obesity, the decreased activity of the ER Ca^2+^ pump sarco/endoplasmic reticulum Ca^2+^-ATPase (SERCA) in hepatocytes leads to ER stress and induces apoptosis [257]. Moreover, impaired Ca^2+^ homeostasis in the ER induces mitochondria dysfunction [226].

Other studies have provided evidence for the involvement of the Ca^2+^ flux in oxidative stress and ER stress induction. In Buffalo rat liver (BRL-3A) cells exposed to a high FFA concentration to mimic NAFLD, induction of oxidative stress stimulates transcription and translation of the protein calcium release-activated calcium channel protein 1 (Orai1), a plasma membrane protein involved in the Ca^2+^ cytosolic flux. The resulting increase in Ca^2+^ entry into the cell activates nuclear factor-kappa B (NF-κB) signaling, which further exacerbates the Ca^2+^ flux, mitochondrial impairment, and ROS generation, enhancing ER stress. Other mechanisms activated by lipid accumulation and Ca^2+^ homeostasis impairment in the liver also enhance ER stress in NAFLD. For instance, in a mouse model of MCD-induced steatohepatitis, increased expression and activation of liver protein kinase Cδ (PKCδ) are associated with ER stress activation. Similar results have been obtained in vitro in mouse hepatic cells cultivated in a palmitic acid-enriched MCD medium. In addition, PKCδ deficiency reduces MCD- and palmitic acid-induced ER stress activation, hepatic TG accumulation, and cell death [259]. These effects may involve CHOP, because in a human hepatic cell line (L02) exposed to palmitic acid, PKCδ deficiency reduces ER stress through decreased *CHOP* expression. In this model, PKCδ silencing also increases SERCA activity, which improves Ca^2+^ homeostasis [260]. In HFD-fed rats, increased plasma FFA is associated with increased expression of markers of oxidative stress and ER stress, as well as Orai1 and NF-κB [261]. Mitochondria and peroxisome functional alterations also can induce ER stress. For example, an imbalance in redox signaling in peroxisomes upon catalase inhibition induces FA accumulation, which leads to ROS production and ER stress [237] (Figure 5).

To summarize, ER stress induced by lipid accumulation during NAFLD promotes ER ROS production and calcium homeostasis disruption that both act to increase oxidative stress during disease progression (Figure 5). ER stress also modulates expression of proteins involved in lipid and glucose metabolism, thus contributing directly to hepatic lipid accumulation [262]. Finally, in addition to mitochondria and ER, the peroxisomes, which are important in H_2_O_2_ production by peroxisomal FA oxidation, contribute to oxidative stress in NAFLD (Figure 3) [226,237,238,263].

#### 3.2.3. Cytoplasmic Production of ROS

Patients with NAFLD present with altered expression of cytochrome P (CYP)450 enzymes, which are involved in detoxification, FA oxidation, inflammation, and oxidative stress (Figure 6) [264]. In these patients, for example, increased serum lipid peroxidation correlates with a higher level of CYP4A11. In contrast, in the hepatocyte cell line HepG2 exposed to FFA, CYP4A11 inhibition led to reduced ROS production associated with decreased lipid accumulation and reduced pro-inflammatory cytokine levels, such as tumor necrosis factor α (TNF-α), interleukin (IL)-6, and IL-1β [265]. In HFD-fed mice, *CYP4A* deficiency reduces hepatic ER stress, apoptosis, insulin resistance, and steatosis, suggesting a role for CYP4A in NAFLD pathogenesis. CYP2E1, which is expressed in ER, mitochondria, and cytosol, is increased in experimental models of NAFLD and in patients with NASH, resulting in ROS generation and inactivation of SOD and catalase [236,266]. CYP2E1 has a high NOX activity that promotes lipid peroxidation [265,266]. Furthermore, overexpression of *CYP2E1* in mice is associated with severe steatohepatitis and upregulation of antioxidant enzymes (SOD2, catalase, and GPX) [267]. CYP1A1 induces ROS production by the reduction of O_2_ to H_2_O_2_ and O_2_^.−^. It also has a role in ω-hydroxylation of PUFAs. The expression of *CYP1A1* is enhanced in oleic acid-stimulated HepG2 cells. In these same cells, CYP1A1 small interfering RNA inhibits lipid peroxidation, whereas overexpression of *CYP1A1* stimulates lipid peroxidation and reduces SOD expression. Collectively, these observations unveil a regulatory role for CYP1A1 in hepatic lipid peroxidation [264].

CYP450 epoxygenase-derived epoxyeicosatrienoic acids (EETs) are a class of lipid mediators that are abundant in liver and that mediate cytoprotective and anti-inflammatory properties. A CYP450-induced EET increase in HFD-fed mice protects against NAFLD progression. Moreover, overexpression of the *CYP2J2* leads to higher EET levels in serum that are associated with decreased hepatic TG levels, inhibition of the NF-κB/JNK signaling pathway, and increased antioxidant enzyme levels. Treatment of HepG2 cells with EET protects against palmitic acid-induced lipotoxicity, oxidative stress, and inflammation [268].

In cytosol, ROS are also produced by many enzymes including xanthine oxidase (XO), cyclo-oxygenase lipoxygenase, and NOX (Figure 6) [226,269,270]. NOX-mediated H_2_O_2_ production causes liver damage [271] and induces a pro-inflammatory response [269]. Patients with NAFLD have a higher NOX2 activity and increased NOX2-derived peptide, a marker of systemic NOX activation [271,272]. In one cohort of children with NAFLD, NOX2 activity increased in parallel with disease severity [271]. Furthermore, HFD-fed mice deficient in *NOX2* develop steatosis but not NASH because of reduced OXPHOS dysfunction [273]. In agreement with this finding, NOX2-induced cellular oxidative stress causes inhibition of OXPHOS activity induced by FFA treatment [270]. Modulation of *NOX* expression also has been reported in experimental models. HFD-fed mice fed have increased *NOX* expression and activity [273], and *NOX1* expression is increased in a mouse model of obesity [274], leading to ROS generation associated with nitrotyrosine protein expression that counteracted oxidative damage [275]. NOX1 also induces ROS production in a mouse model of steatosis [274].

In humans, *NOX4* mRNA expression is upregulated in the liver of patients with NASH [275,276]. The *NOX4* single nucleotide polymorphism rs3017887 is associated with increased ALT levels in liver biopsies from patients with NAFLD, indicating hepatocyte damage [269]. In a mouse model of diet-induced NAFLD, hepatocyte-specific deletion of *NOX4* or its pharmacological inhibition reduces oxidative stress and fibrosis. Cell culture findings show that NOX4 promotes oxidative stress by ER stress activation [276]. In summary, increased NOX isoform expression and activity are associated with NAFLD and lead to oxidative stress induction via ROS generation concomitant with activation of ER and mitochondrial stress (Figure 4 and Figure 5).

Oxidized phospholipids (OxPLs) play a role in oxidative stress induction through generation of cellular ROS, which promote fibrosis and inflammation that result in NASH progression. OxPLs are produced upon overnutrition or in patients with NAFLD. Treatment of primary hepatocytes from *Ldlr*^−/−^ hyperlipidemic mice with an OxPL mixture promotes ROS accumulation in the cytoplasm and all organelles, resulting in mitochondrial dysfunction [277]. Treatment of the immortalized human hepatic stellate cell line with OxPLs stimulates fibrogenic gene expression [278]. Furthermore, in a mouse model of NASH, the neutralization of OxPL decreases inflammatory mechanisms [277]. Toll-like receptor *(TLR)-4* and *TLR-2* are expressed in Kupffer cells and stellate cells, which are associated with fibrosis. Mice fed an HFD supplemented with lecinoxoids (synthetic OxPLs), which inhibit these TLRs, show reduced liver fibrosis and inflammation (reduced IL-1B and IL-6) [279]. Furthermore, the neutralization of OxPL in a mouse model of NAFLD inhibits progression to HCC [277].

Iron metabolism, which is modified in patients with NAFLD [75,280,281,282], leads to ROS generation because of the ability of the ferrous iron to catalyze the production of OH^−^ from H_2_O_2_, known as the Fenton reaction [257] (Figure 2, Figure 3 and Figure 4). This point is further developed below in Section 5.

#### 3.2.4. Gut Microbiota and Liver Oxidative Stress

Mice lacking the gut microbiota (germ-free mice) display an altered hepatic pool of glutathione [283]. The gut microbes also alter levels of amino acids and N-acetylated amino acids circulating in the mouse portal vein, affecting host amino acid and glutathione metabolism in the intestine and the liver [284]. A recent study showed that the gut microbiome induces the Nrf2 antioxidant response pathway in the liver [285] and that exogenous administration of Lactobacilli could amplify this response and protect against oxidative liver injury. Of great interest, these authors identified a Lactobacilli-derived metabolite (namely 5-methoxyindoleacetic acid) that could activate hepatic Nrf2 and in part mediate these beneficial effects. The relevance of this gut–liver mechanism in humans remains to be studied.

Unhealthy eating modifies the gut microbiota, as others have reviewed [286,287]. This dysbiosis participates in the development of metabolic diseases and plays a major role in NAFLD pathogenesis [288,289]. In fact, the development of NAFLD correlates with (i) dysbiosis; (ii) a leaky intestinal barrier; (iii) impaired mucosal immunity; (iv) bacteria and bacterial components reaching the liver through the portal vein, as well the bacterial metabolites such as lipopolysaccharides, trimethylamine-N-oxide, N,N,N-trimethyl-5-aminovaleric acid, and endogenous ethanol; and (v) impaired bile acid homeostasis [286,287]. These alterations participate in increased hepatic inflammation and oxidative stress and promote NAFLD development and progression to NASH.

Altogether, whereas the role of oxidative stress in NAFLD pathogenesis has been demonstrated in both clinical and animal studies, the underlying mechanisms are complex and not completely understood. Alterations in mitochondrial respiratory complex activity because of lipid accumulation in NAFLD make the mitochondria the main source of ROS in this condition. ER stress, cytochromes, NOX, and OxPLs are also key components in ROS generation. The consequences of ROS production in NAFLD are enhanced oxidative stress and altered lipid metabolism, which further promote lipid accumulation. In addition, increased ROS production may modulate insulin signaling and inflammatory processes, thus promoting insulin resistance and inflammation, which are important features of NAFLD progression [226] (Figure 2); however, the specific mechanisms underlying oxidative stress–promoted NAFLD need further study. Furthermore, several environmental factors, such as the nutrients and contaminants discussed below, affect oxidative stress in NAFLD.

## 4. Oxidative Stress in NAFLD: Role of Nutrients

An unhealthy lifestyle is strongly associated with NAFLD. Of note, dietary constituents are key factors in NAFLD pathogenesis. Such factors include a high intake of calories and an excessive consumption of saturated fats, refined carbohydrates, and animal proteins. Several nutrients are well known to affect the metabolic pathways leading to hepatic fat accumulation in the first steps of NAFLD development. Accumulating data also support the idea that diet-mediated alterations include increased ROS production and oxidative stress, which are key factors in the progression of NAFLD to NASH, as noted above (Figure 7). Given the essential role of nutrition in the etiology of NAFLD, several epidemiological studies have evaluated its association with dietary habits.

### 4.1. Dietary Oxidants in NAFLD

A correlation between consumption of different nutrients and oxidative stress has been previously shown in patients with NASH. Intake of saturated fat is associated with a lower ratio of reduced to oxidized glutathione (GSH/GSSG), suggesting a pro-oxidant effect. In contrast, consumption of fiber, MUFAs, PUFAs, vitamin E, selenium, and folate protect against oxidative stress [290].

HFDs induce obesity, insulin resistance, and hepatic steatosis. As noted above, an excess of lipids in hepatocytes is associated with lipotoxicity and mitochondrial overload, resulting in impaired FA catabolism and ROS production. Increased exogenous FA delivery by liver perfusion or by HFD increases mitochondrial oxidative metabolism, resulting in hepatic oxidative stress during NAFLD in mice [248]. Of interest, FA storage as TG, which avoids accumulation of hepatic FFAs, protects against hepatic lipotoxicity and oxidative stress [291]. In line with this observation, the level of TG and free cholesterol increases in the liver in patients with NAFLD, whereas the level of FFAs is unaltered [292].

A limited number of studies have assessed the relationship between dietary saturated FA intake and oxidative stress. Medium-chain saturated FAs have shown antioxidant effects in rats through reduced lipid peroxidation and increased activity of antioxidant enzymes [293,294]. In contrast, saturated FAs in the steatotic rat liver induce hepatocyte ER stress and apoptosis [295]. In elderly Japanese patients, a correlation has been reported between high consumption of saturated FAs, particularly short-chain saturated FAs and medium-chain saturated FAs, and decreased urinary 8-hydroxy-2’-deoxyguanosine (an oxidative stress marker) and ameliorated hypertension [296]. Overall, the effect of saturated FAs on oxidative stress in NAFLD remains unclear and needs further investigation. Differences in FA chain lengths may contribute to differences in their effects on oxidative stress: short-chain saturated FAs and medium-chain saturated FAs may be antioxidant, whereas long-chain saturated FAs increase oxidative stress.

In addition, a high ratio of omega-6 to omega-3 FAs reduces mitochondrial respiratory functions and increases ROS levels in human hepatoma cells. Decreasing this ratio prevents diet-induced NASH and hepatic oxidative stress in rats [297,298].

FAs are not the only compounds that accumulate in the mitochondria. Cholesterol is consistently elevated in human and mouse fibrotic NASH, and its mechanistic link to NASH development has been explored recently [59]. In hepatocytes from a NAFLD rat model, mitochondrial cholesterol accumulation decreases mitochondrial glutathione, which is crucial for scavenging ROS produced during mitochondrial metabolism, and sensitizes mitochondria to inflammatory cytokines [299]. Mice fed a high-cholesterol diet show a modified transcriptome, including modulation of several pathways involved in cell death and oxidative stress. Cholesterol accumulation increases ROS generation, leads to expression of antioxidant genes such as *SOD2*, and modifies expression of genes regulating the glutathione pathway (decreased *GPx-1*, increased *GPx-4*) [300]. In addition, oxidation of free cholesterol generates oxysterols, which are elevated in NAFLD and may participate in lipid synthesis and inflammation [301]. In patients with NASH, cholesterol accumulation decreases mitochondrial membrane permeability and induces defective mitochondrial GSH transport from cytosol to the mitochondria [256].

Interactions among different lipid classes may play a role in oxidative stress induction. For example, dietary fat and cholesterol act in synergy to promote NAFLD progression through impairment of mitochondrial function and biogenesis. Mice fed a high-fat, high-cholesterol diet develop more severe liver lesions, including inflammation and fibrosis, and show mitochondrial dysfunction such as altered respiration, increased H_2_O_2_ production, and impaired ATP homeostasis, compared to each diet (high fat or high cholesterol) fed separately [302,303]. The combination of dietary cholesterol with PUFAs leads to high levels of oxidized peroxiredoxins and protein carbonyls, indicating severe hepatic oxidative stress [304]. Of interest, a ketogenic diet increases rat liver antioxidant capacity [305].

In the liver, fructose is mainly metabolized by fructokinase, which uses ATP to phosphorylate fructose into fructose-1-phosphate. Thus, fructose metabolism decreases ATP levels, resulting in oxidative stress induction and uric acid generation, which also has pro-oxidative effects [50,306]. Fructose drinking in rodents also induces gut microbiota changes and CYP2E1-dependent intestinal oxidative stress, which both lead to endotoxemia and increased severity of steatohepatitis with fibrosis [307]. Rats fed a diet rich in fructose for 3 weeks show decreased catalase activity and GSH levels in the liver, whereas the level of NOX and the content of protein carbonyl groups, a marker of oxidative stress, are increased [308,309]. These effects are associated with changes in carbohydrate metabolism and increased hepatic TGs, suggesting a switch towards lipid synthesis rather than mitochondrial oxidation, which may protect against ROS production [309]. A fructose-rich diet also causes oxidative damage to hepatic lipids, proteins, and mitochondrial DNA, as shown by increased lipid peroxidation and nitrotyrosine levels in the liver and higher levels of plasma 8-hydroxydeoxyguanosine generated by oxidative damage to DNA. In addition, administration of the antioxidant lipoic acid or an inhibitor of NOX protects against fructose-induced hepatic insulin resistance, oxidative stress, and inflammation, reinforcing the relationship between hepatic oxidative stress and high fructose intake [308,310]. Fructose intake also enhances hepatic oxidative stress of maternal HFD offspring because of decline in the antioxidant defense system [311].

In patients with NASH and hyperferritinemia, iron reduction by a combination of iron-restricted diet and phlebotomy reduces serum iron and ferritin levels, liver enzymes, and hepatic oxidative damage to DNA [312]. In preclinical models, iron excess leads to hepatic TG accumulation, but the effect on oxidative stress is controversial. For example, dietary iron supplementation enhances HFD-induced hepatic steatosis and inflammation, whereas no change in hepatic oxidative stress is observed [313]. In contrast, iron supplementation in genetically obese mice causes hepatocellular ballooning and hepatic oxidative stress by increasing lipid peroxidation and decreasing expression of antioxidant genes [314]. A recent study indicated that the hepatocyte-specific deletion of the iron chaperone, poly r(C) binding protein 1 (*PCBP1*), results in liver iron depletion, hepatic steatosis and inflammation, increased lipid peroxidation, and activation of the Nrf2 and glutathione pathways. Both an iron-restricted diet and treatment with vitamin E can prevent steatosis and hepatocyte damage associated with PCBP1 deficiency, suggesting that iron mediates its toxic effects in part through the production of ROS [315]. Liver biopsies of 222 patients (men and women) with NAFLD showed that serum ferritin levels, which are biomarkers of iron metabolism, are positively associated with a higher NAFLD activity score [316]. The relation between ferritin level and NAFLD is mostly significant in men and associated with hepatic iron deposition, fibrosis, and NASH [77,317]. A correlation between ferritin level and NAFLD development also has been observed in a cohort of postmenopausal women [317].

Collectively, observations from dietary interventions show that cholesterol and fructose promote oxidative stress in NAFLD pathogenesis. Meat processing of meat also causes protein oxidation, which may be associated with oxidative stress [318]. In addition, the interactions among different types of lipids may play a role in the induction of oxidative stress during disease development. The pro-oxidant effects of saturated lipids, animal-based proteins, and iron are not well understood and need further investigation in experimental and large-scale clinical studies. Although excess of certain nutrients is clearly associated with oxidative stress induction in NAFLD, other nutrients have antioxidant properties. Below, we discuss the antioxidant role of these nutrients in preventing NAFLD development.

### 4.2. Dietary Antioxidants in NAFLD

Several meta-analyses have demonstrated the effectiveness of omega-3 PUFA supplementation in the dietary management of NAFLD, especially for liver fat content, although a detailed mechanism of its effect on liver inflammation and fibrosis is still unclear [319,320,321,322,323]. Omega-3 PUFAs reduce liver TG accumulation mainly through regulation of hepatic lipid metabolism and inflammation [324]. In experimental studies, supplementation with omega-3 PUFAs, such as eicosapentaenoic acid (EPA) and docosahexaenoic acid (DHA), reduces HFD-induced steatosis in part through induction of antioxidant responses [325]. Furthermore, omega-3 PUFA supplementation reduces hepatic oxidative stress in HFD-fed mice, as shown by a decreased GSSG/GSH ratio and protein carbonylation content in the liver [326]. Omega-3 PUFA supplementation also reduces the WD-induced NOX pathway [327] and can decrease oxidative stress through indirect mechanisms such as Nrf2 upregulation because the oxidation products of DHA and EPA are potent Nrf2 activators [328]. Dietary DHA is more efficient than EPA in reducing WD-induced NASH and hepatic oxidative stress [327,329]. In humans, the effects of PUFAs on oxidative stress in NAFLD are less clear. According to a recent systematic review of clinical trials, the effect of omega-3 PUFAs on oxidative stress in human NAFLD is inconclusive and varies from study to study depending on the oxidative stress marker measured [323].

MUFAs are the main FAs found in extra-virgin olive oil, which is associated with the prevention of diet-induced liver oxidative stress [330]. However, a specific mechanism is unknown, and antioxidant compounds such as phenols, which are present in high quantity in extra-virgin olive oil, are more likely responsible for these beneficial effects [331].

A few dietary amino acids affect NAFLD pathogenesis in several ways, including gut epithelium integrity, inflammation, fibrosis, and glucose homeostasis. We will only briefly address amino acids here because a recent review has described their effects in some detail [332]. Branched-chain amino acids repress inflammation, apoptosis, and fibrosis by downregulation of the transforming growth factor-β and Wnt/β-catenin pathway, and steatosis and mitochondrial dysfunction by downregulating FA synthase and upregulating mTOR. Glutamine attenuates inflammation by upregulating PPARγ. The α-amino acids citrulline and arginine downregulate TLR-4, TNFα, IL-6, and endotoxins, which ameliorates inflammation and apoptosis, and serine increases AMPK activity to repress steatosis and mitochondrial dysfunction. Serine is also a GSH precursor, and its supplementation has been reported to decrease hepatic steatosis in a proof-of-concept human study [333]. Of interest, glycine supplementation increases thermogenic activity in hepatic mitochondria, mitigates liver steatosis, improves insulin sensitivity, and reduces serum lipid levels by repressing p38, JNK, and TLR-4, and increasing glucagon-like peptide-1/glucagon [332]. Methionine also has an important hepatic impact because compromised levels in methyl-group donors result in alterations promoting the development of NAFLD in animal models and humans. In mice, both dietary methionine deficiency and high methionine supplementation induce anomalies associated with NAFLD progression, including an effect on lipid metabolism and one-carbon metabolic pathways disturbances. Oxidative and ER stress is also increased [334].

Choline, which is not an amino acid, is also an essential methyl donor, and its deficiency results in impaired VLDL secretion and hepatic fat accumulation and liver damage, including impaired mitochondrial function. The result is reduced FA oxidation and stimulated production of ROS with downstream deleterious effects. Of interest, the MCD diet has become a classical diet for triggering NASH. The absence of two main ingredients, methionine and choline, alters mitochondrial β-oxidation and VLDL synthesis [335], causing relatively rapid fatty liver development, together with oxidative stress and modifications in NASH-promoting adipokine and cytokine profiles, contributing to liver injury [336]. MCD-fed rodents undergo significant weight loss and present with decreased serum TG, cholesterol, glucose, insulin, and leptin [337]. These effects are a limitation of this diet model because the metabolic profile it induces is very different from that of human NASH, which must be considered when using the MCD in studies [338].

Taurine is a sulfur-containing β-amino acid that is not used for protein synthesis. The main sources of taurine are fish, meat, and dairy. It has several health benefits, and people also take it as a supplement to improve physical performance. In a mouse NAFLD model, taurine treatment alleviated HFD-induced reduction of catalase and SOD activity, and, in HepG2 cells, it suppresses FA-induced lipid accumulation and reduces ROS production in concert with the FA-induced alteration in mitochondrial membrane potential. These findings supporting an antioxidative effect of taurine show that it can attenuate NAFLD progression in animals [339].

Micronutrients are required in microgram or milligram quantities for physiological functions, such as cellular metabolism and tissue function [340]. These compounds, including vitamins, minerals, and phytochemicals, are also important in NAFLD pathogenesis [72,341], mainly through modulation of anti-inflammatory and antioxidant pathways. Liver plays an essential role in micronutrient metabolism by contributing to their uptake, transport, and storage, and by producing binding, transport, and regulatory proteins required for their action at the cellular level [72]. Micronutrients with antioxidant capacity include vitamin D, vitamin E, and phytochemicals such as carotenoids and polyphenols.

Vitamin D is often deficient in liver diseases and is associated with NAFLD and NASH in epidemiological studies. In fact, the anti-inflammatory, antifibrotic, and antiproliferative effects of vitamin D on NAFLD and NASH have been documented in both human cohorts and animal models [342,343,344,345]. Preclinical models have provided critical information for designing possible related therapeutic strategies for the treatment of NAFLD/NASH [342].

Vitamin E supplementation in adults with NAFLD does not improve fibrosis but ameliorates hepatic steatosis and inflammation [346]. Animal studies show that vitamin E improves NAFLD/NASH by repressing oxidative stress and inflammation [347]. We have recently reviewed the impact of vitamin E on NAFLD/NASH more extensively [348].

Natural polyphenols found in vegetables, fruits, coffee, and wine are a class of phytochemicals that include several different compounds sharing a common phenolic structure. They are classified as flavonoids and non-flavonoids. Their beneficial effects mainly trace to their antioxidant properties, and they also influence glucose and lipid metabolism and exert anti-inflammatory, antifibrogenic, and antitumoral effects [349]. In addition, they reduce de novo lipogenesis by regulating the activity of SREBP-1c and stimulate FA β-oxidation possibly via activation of AMPK [350]. Furthermore, several compounds of the polyphenol family, including green tea catechins, curcumin, resveratrol, and quercetin, appear to reduce liver enzymes, lipid peroxidation, and inflammation markers [342]. Carotenoids also have beneficial effects on the liver, mainly through their antioxidant capacity and by regulating expression of genes involved in inflammation and lipid metabolism [351]. Furthermore, many other natural products present in the diet confer hepatoprotective effects by mechanisms related to improved gut dysbiosis and ameliorated intestinal barrier permeability, and inhibiting steatosis by mechanisms acting on inflammation, oxidative stress, fibrosis, and apoptosis [286].

### 4.3. Gut Microbiota and Dietary Antioxidants in NAFLD

Natural dietary supplements can support microbiota homeostasis or improve it when it is disturbed [286,287]. The use of probiotics (*Lactobacillus* and *Bifidobacterium*) can ameliorate NAFLD [352,353,354,355]. Similarly, liver health can be supported by well-planned consumption of several classes of food-derived compounds that are possibly involved in microbiota maintenance. Among these are functional oligosaccharides (fructo-oligosaccharides, galacto-oligosaccharides, and chitosan oligosaccharides) [275,356,357]. Dietary fiber is another, catabolized through fermentation by the lower gastrointestinal tract microbiota to short-chain FAs comprising butyrate, propionate, and acetate that ameliorate NAFLD pathogenesis [358,359]. The microbiota can convert functional amino acids, such as L-tryptophan, to indoles and its derivatives, which improve gut and liver health. In preclinical studies, oral administration of another amino acid, L-glutamine, has been proved protective against diet-induced NASH progression [287,360]. Carotenoids improve the gut barrier and immune homeostasis and, as noted, are known for their anti-inflammatory and antioxidant properties [361]. Polyphenols repress insulin resistance, oxidative stress, and inflammation and stimulate FA β-oxidation, which collectively confer a protective effect against NAFLD. The ω-3 PUFAs have a broad range of beneficial effects in preclinical and clinical studies in preventing or treating NAFLD. They improve intestinal barrier integrity and thus reduce bacterial translocation, ameliorate bile acid homeostasis, and repress liver inflammation via activation of the nuclear farnesoid X receptor [287].

These effects of various compounds in interaction with the gut microbiota indicate that nutritional interventions offer promise in NAFLD/NASH prevention and treatment and that their implementation will evolve toward personalized dietary therapies with consideration of each patient’s genomic, metabolic, and microbiotic profile. The doses of individual micronutrients appear to be important, with some doses offering benefits and others resulting in side effects [72]. In addition, oxidative stress is an important factor in cancer initiation but may also offer some beneficial toxicity against cancer cells. Thus, antioxidants could have harmful effects in patients with HCC, for instance. More studies investigating the long-term effects and outcomes in patients with advanced NASH and liver cancer are needed [362].

## 5. Oxidative Stress in NAFLD: Role of Pesticides

Insecticides, herbicides, and fungicides have shown pro-oxidative properties in various organs and tissues, including the liver (Table 3). Among the insecticides, neonicotinoid, pyrethroid, organophosphorus, and organochlorine compounds are the most often studied. Neonicotinoid-associated effects in various organs include the production of ROS [183], and reactive nitrogen species (RNS) [363]. Studies assessing the role of oxidative stress in permethrin- and deltamethrin-induced toxicity in animal models concluded that the toxicity of these compounds is mainly the result of their pro-oxidative properties. Consistent with this finding, natural antioxidants diminish deltamethrin-induced oxidative stress damage [364,365]. Permethrin-mediated ER stress was recently reported in hepatic cell lines [366]. Other pyrethroid pesticides also exhibit pro-oxidative properties in the liver, as shown in Table 3 [367,368,369,370,371], and various organophosphorus and organochlorine insecticides cause oxidative stress in animal liver [136,183,371,372]. Furthermore, several studies have reported the pro-oxidative properties of some herbicides [151,153,154,221,373,374,375,376] and chemical families of fungicides (carbamate, benzimidazole, strobilurin, azoles, triazoles [159,377,378,379,380,381,382,383]) (Table 3). Liver oxidative stress is also reported in animals exposed to a mixture of pesticides [168,193,384] (Table 1 and Table 3). Collectively, these data underscore the pro-oxidative properties of insecticides, herbicides, and fungicides.

ROS can be produced from several sources during metabolism, as a result of biotransformation reactions of pesticides (herbicides, insecticides, fungicides, miticides) by CYP450 enzymes, which catalyze the oxidation reactions [192,385,386,387]. For example, oxidative stress generated by exposure to the insecticide permethrin is linked to the detoxifying activities of CYP450 enzymes, and other enzymes involved in oxidation and hydrolysis processes [364]. Organochlorine compounds such as DDT lead to an increased ROS production through activation of *CYP450* gene expression involved in its detoxifying pathway in exposed rat liver [211]. The biotransformation reactions of organophosphorus compounds, during which an excess of free radicals is generated, also produces ROS [187].

**Table 3 biomolecules-10-01702-t003:** Impact of pesticide exposure on oxidative stats in in vivo models. Acceptable Daily Intake (ADI) values were from https://ephy.anses.fr/ and https://ec.europa.eu/food/plant/pesticides/eu-pesticides-database/. ALT, alanine aminotransferase; AST, aspartate aminotransferase; BW, body weight; CYP, cytochrome P450; EROD, ethoxyresorufin-O-deethylase; MDA, malondialdehyde; GPX, glutathione peroxidase; GSH, reduced glutathione; GSSG, oxidized glutathione; GST, glutathione-S-transferase; IL-1 IL-6, Interleukin 1 and 6; ROS, reactive oxygen species; SIRT, sirtuin; SOD, superoxide dismutase; TNFα, Tumor necrosis factor α.

Type of Pesticide	Chemical Family	Active Substances (ADI mg/kg BW/day)	Experimental Model	Oxidative Impacts	Refs.
Insecticide	Pyrethroid	B cypermethrin enantiomers (0.0016)	Zebrafish Pesticide in water at 0.1 to 4 µg/L for 4 days	Changes in MDA content and in antioxidant enzyme activities for some, but not all enantiomers	[367]
			Male mice Daily oral gavage at 5 mg/kg BW for 2, 4, or 6 weeks	Increased hepatic ROS, GSH, and MDA levels (1S-cis enantiomer) Changes in catalase and, to a lesser extent, SOD activities	[368]
		Bifenthrin (0.015)	Male and female rats Daily gavage with 5.8 mg/kg BW for 20 or 30 days	Increased liver MDA content Decreased liver antioxidant enzymes activity (at 30 days)	[369]
			Male and female mice Daily intraperitoneal injection of 2, 4, or 8 mg/kg BW for 7 days	Increased MDA and ROS levels, and SOD activity at 4 or 8 mg/kg BW	[370]
Insecticide	Pyrethroid	Lambda-cyhalothrin (0.0025)	Male rats Daily oral gavage with 1, 2, 4, or 8 mg/kg for 6 consecutive days	Increased total hepatic CYP content and EROD activities Decreased GST and GPX and increased SOD 2 enzyme activities Increased expression of *CYP* genes and genes involved in inflammatory processes, oxidative stress, and apoptosis No change in ROS levels	[138]
Insecticide	Pyrethroid or organophosphorus	Deltamethrin (DTM) (0.01) Or Chlorpyrifos (CPF) (0.001)	Male rats fed a pesticide-enriched diet for 16 weeks containing either to chlorpyrifos alone (1 or 15 mg/kg food/day) or deltamethrin alone (5 or 35 mg/kg food/day) or the mixture of chlorpyrifos and deltamethrin at 1 mg/kg/day or 5 mg/kg/day	Increased hepatic MDA content Changes in antioxidant enzyme activities in rat fed CPF- or DTM- or mixture-enriched diet	[371]
Herbicide	Glycine derivate	Glyphosate (0.5)	Male rats Intraperitoneal injection every 2 days of 50 mg/kg BW in the presence of absence of 20 mg/kg BW quercetin for 15 days	Increased liver MDA content Increased H_2_O_2_ generation Decreased liver metallothionein content Decreased enzymatic antioxidant activity and non-enzymatic component level Effects were suppressed in quercetin co-treated animals	[373]
			Male rats Oral daily gavage with 5, 50, or 500 mg/kg BW for 25 days	Increased serum MDA level at 50 mg/kg BW Decreased hepatic SOD activity and increased H_2_O_2_ production at the highest assessed dose Increased liver *IL-1*, *IL-6*, *TNF-alpha*, *GPX2*, *caspase*, and *SIRT1* mRNA levels	[153]
			Male rats Oral exposure to Roundup through drinking water (0.05 µg/L glyphosate) for 2 years	Induction of oxidative stress (glutathione metabolism) determined through analysis of liver proteome and metabolome	[151]
Herbicide	Glycine derivate	Glyphosate (0.5)	*Caenorhabditis elegans* A 30-min exposure to 2.7%, 5.5%, or 9.8% glyphosate formulation (TouchDown containing 52.3% glyphosate)	Inhibition of the mitochondrial electron transport chain Inhibition of proton gradient Inhibition of ATP and H_2_O_2_ production Upregulation of *GST4* gene expression	[374]
	Triazine	Atrazine (0.02)	Male mice Daily intraperitoneal injection of 100 to 200 mg/kg BW for 1 week	Decreased BW Slight but significant increase in hepatic SOD activity at the highest dose	[221]
	Aryloxy acid	2,4 dichlorophenoxyacetic acid (2,4 D) (0.02)	Male rats Daily oral gavage with 15, 75, or 150 mg/kg BW for 28 days	Increased liver weight Increased serum levels of TG and low-density lipoprotein cholesterol Increased levels of saturated fatty acid and decreased levels of unsaturated fatty acid Changes in antioxidant enzyme activities	[375]
	Dinitro-aniline	Trifluralin (0.015)	Male rat isolated mitochondria Exposure to 1 to 100 µM	Impairment of oxidative phosphorylation Mitochondrial swelling No change in ROS production and glutathione levels	[376]
		Pendimethalin (0.125)	Male rats Daily oral gavage at 62.5, 125, or 250 mg/kg BW for 14 days	Decreased liver glutathione content Decreased liver SOD, catalase, and GST activities	[154]
Fungicide	Carbamate	Thiophanate-methyl (0.08)	Male rats Single intraperitoneal injection of 300 to 500 mg/kg BW Observations 3 days later	Increased antioxidant enzyme activity Decreased liver GSH and vitamin C levels	[388]
	Strobilurin	Azoxystrobin (0.2)	Male and female zebrafish Dilution in water at 1, 10, or 100 µg/L 28 days of exposure	Induced liver ROS production and MDA level Changes in antioxidant enzymes activities	[378]
	Strobilurin	Azoxystrobin (0.2) or picoxystrobin (0.043)	Zebrafish fertilized eggs exposed in water to 0.25, 2.5, 25, or 250 mg/L azoxystrobin and 0, 0.02, 0.2, 2, or 20 mg/L picoxystrobin for 24, 48, 72, 96, or 144 h Adult male and female zebrafish exposed for 28 days to the above doses	Significant increased antioxidant enzyme and detoxifying enzyme activities Increased content of MDA in the larval zebrafish study The effect on activities of the antioxidant and detoxification enzymes differed between the sexes in adult zebrafish liver, with male zebrafish more sensitive to azoxystrobin than female zebrafish	[379]
	Triazole	Propiconazole (0.04)	Male rats Daily gavage at the equivalent level of no observed effects level X1, X3, or X5 for 28 days	Changes in antioxidant enzyme activities	[380]
Fungicide	Triazole	Propiconazole	Male mice Pesticide-enriched diet (2500 ppm) or intraperitoneal injections for 4 days	Decreased GSH/GSSG ratio Decreased cytochrome *c* reductase activity Increased protein oxidation	[381]
			Male mice Pesticide-enriched diet at 500, 1250, or 2500 ppm for 4 days	Reduced glutathione levels Alteration in ascorbate metabolism	[382]
		Tebuconazole (0.03)	Male rats Daily oral gavage at 10, 25, or 50 mg/kg peroral for 28 days	Increased hepatic levels of various CYP proteins Increased antioxidant enzyme activities Decreased glutathione content	[383]
	Imidazole	Imazalil (0.025)	Male mice Environmental doses in drinking water for 15 weeks	Increased levels of ROS and MDA Changes in SOD activity Decreased level of GSH in the liver	[159]
Fungicide	Dithiocarbamate	Maneb (0.05)	Female mice Daily intraperitoneal injections to 1/2, 1/4, 1/6, or 1/8 of the LD_50_ for 7 days	BW loss Increased serum AST and ALT activities Decreased serum high-density lipoprotein cholesterol and increased cholesterol and low-density lipoprotein cholesterol Increased liver MDA level and H_2_O_2_ generation Decreased GPX, SOD2, and catalase activity Decreased liver GSH and vitamin C levels	[377]
Insecticide and fungicide	Benzimidazole and organophosphorus	Mixture carbendazim (0.02) and chlorpyrifos (0.001)	Female rats Orally exposed during 7 day to 50 mg/kg BW/day chlorpyrifos and 10 mg/kg BW/day carbendazim	Decreased enzymatic activity and non-enzymatic antioxidant component levels	[384]

Moreover, all mitochondrial complexes, except complex IV, are targets of at least one pesticide family (Table 4). In chloroplasts, herbicides are photosynthesis inhibitors and mediate their actions by inhibiting the electron transport chain. Some functional similarity between plant chloroplasts and mammalian mitochondria may explain the mitochondrial toxicity of the herbicide triazine, which in plants inhibits the photosynthetic process that it targets [147]. Indeed, the triazine herbicide blocks the activity of the oxidative phosphorylation complex I, the NADH dehydrogenase complex, in an animal model (Table 4) [148]. Atrazine inhibits complexes III and V [148,389]. Other herbicides such as Diuron and glyphosate inhibit complex III [390,391]. Organochlorine insecticides affect complexes II and III, which contain the enzyme succinate dehydrogenase, and causes inhibition of succinate translocation, likely explaining insecticide suppression of mitochondrial respiration [392]. Organophosphate insecticides inhibit mitochondrial complex I in birds [393], and effects on complex IV and/or V have been reported in mammalian models [394,395]. Complex V is also inhibited by imidacloprid, abamectin, and DDE [392,396,397].

The strobilurin family of fungicides, known as succinate dehydrogenase inhibitors (SDHIs), inhibits fungus mitochondrial respiration via blockade of electron transfer between cytochrome *b* and cytochrome *c1*, which results in oxidative stress [378]. In addition to the inhibitory effect on succinate dehydrogenase, other SDHI fungicides inhibit complex III (ubiquinol cytochrome *c* reductase) with varying efficiency in human and animal cells [184,381,398,399]. The herbicide glyphosate targets the mitochondrial electron transport chain in duckweed, in particular complex III, resulting in ROS production; however, its impact on mammalian mitochondria is not yet documented [400]. Some pesticides may affect more than one mitochondrial complex [148,395] (Table 4). For example, SDHI fungicides act on mitochondrial complexes I, II, and III (Table 4). In brief, various chemical families of insecticides, except DDE, alter complex V, whereas rotenone and malathion act on complexes I and II and I, IV, and V, respectively. Fungicides affect complexes II and III of the mitochondrial respiratory chain, and herbicides target complexes I, III, and V (Table 4). It is noteworthy that some fungicides (fludioxonil and Maneb) and the herbicide paraquat induce oxidative stress by acting on both mitochondria complexes and cytosolic NADPH oxidase [401,402].

Pesticide exposure affects antioxidant enzyme expression and/or activity and/or levels of antioxidant compounds such as GSH (Table 3). Pesticides can influence antioxidant pathways by modifying the expression of *Nrf2*. In HepG2 cells, DDE exposure decreases Nrf2 protein level and subsequently glutamyl-cysteine synthase activity [403]. Changes in *Nrf2* promoter methylation have been identified in germ cells from adult mice exposed to the herbicide methyl-parathion for 5 days [404]. The exposure of rats to the pyrethroid insecticide bifenthrin for 60 days at a high dose led to an induction in gene expression, including of *Nrf2*, in the hippocampus [405], and the fungicide prochloraz induced Nrf2 activity in a gene reporter assay [406]. Exposure to pesticides such as atrazine, chlorpyrifos, zinc dimethyldithiocarbamate, vinclozolin, paraquat, and rotenone alters Sirt expression (mainly Sirt1, 3, and 6)—all members of a NAD^+^-dependent type III deacetylase enzyme family considered to operate as stress adaptors under oxidative, genotoxic, and metabolic stresses [407,408,409,410,411,412,413]. The mitochondrial scavenger of ROS, UCP-2, is also induced by the organochlorine pesticide DDE [203]. Pesticides induce ER stress in association with the development of some pathologies but without metabolic disturbance [414]. Additionally, the antiparasitic niclosamide induces ROS generation that enhances ATF3 and CHOP expression and ER stress in hepatoma HepG2 cells [415]. However, the pyrethroid insecticide permethrin induces ROS production without inducing ER stress in HepG2 cells [366]. Altogether, these findings indicate that pesticides affect several processes that result in oxidative stress: the CYP450 system, mitochondrial respiration, antioxidant pathways, the stress adaptor system, and scavenger ROS activity. 

**Table 4 biomolecules-10-01702-t004:** Impact of pesticide on mitochondrial electron transport chain complexes.

Type of Pesticide	Active Molecule	Electron Transport Chain Complexes					Model	References
		I NADH-Coenzyme Q Reductase	II Succinate-Coenzyme Q Reductase	III Reduced Coenzyme Q-Cytochrome C Reductase	IV Cytochrome C Oxidase	V F0F1 ATPase		
Insecticide	Rotenone	x					Isolated mitochondria from exposed HL-60 cells	[405]
	DDT/DDE		x	x		x	Rat liver or heavy beef heart isolated mitochondria or sonicated submitochondria particles	[392]
	Monocrotophos					x	Rat muscle mitochondria isolated from exposed animals	[394]
	Imidacloprid					x	Rat liver isolated mitochondria	[397]
	Malathion	x			x	x	Rat muscle mitochondria isolated from exposed animals	[395]
Herbicide	Atrazine					x	Rat liver, isolated mitochondria	[389]
		x		x			Rat muscle mitochondria isolated from exposed animals	[148]
	Diuron			x			Rat liver, isolated mitochondria	[390]
	Glyphosate			x			Duckweed	[391]
Fungicide	Propiconazole		x	x			Exposed mouse whole liver	[381]
	Azoxystrobin			x			Rat liver, isolated mitochondria	[184]
	Manzate		x				Exposed Caenorhabditis elegans	[398]
Insecticide	4 Succinate dehydrogenase inhibitors		x	x			Exposed HEK cells	[399]
Antiparasitic	Abamectin					x	Rat liver, isolated mitochondria	[396]

Besides oxidative stress, pesticide exposure may lead to NAFLD through their impact on lipid metabolism by (i) modifying fatty acid uptake and efflux [171] and modulating FFA transport as described for insecticides and herbicides, (ii) increasing lipogenesis ([173,406] (Table 1), (iii) altering oxidation pathways [175]; insecticides, herbicides, and fungicides affect both β-oxidation and lipogenesis (Table 1 and Table 2), (iv) interacting with nuclear receptors [416,417,418,419,420,421] involved in the control of metabolism as demonstrated for pyrethroid and neonicotinoid insecticides [176,177,178], and imidazole or triazole fungicides [158,382] or a pesticide mixture [193], but implication in NAFLD is not proven.

Some pesticides may ameliorate lipid metabolism. For example, environmental doses of Boscalid, a widely used fungicide, induce gene expression that promotes β-oxidation and inhibits lipogenesis in the zebrafish liver [179]. Similarly, oral exposure to the carbamate fungicide propamocarb in mice during a 4-week period leads to a down regulation of genes involved in glycolysis, TG, and fatty acid synthesis in the liver [180]. Pesticide alter glucose metabolism through activation of glucose uptake, glycogenolysis, gluconeogenesis (as reviewed for insecticides in [136]) or through inhibition of the mitochondrial respiratory complex (Table 4), and modulation of *Chrebp* gene expression levels [185], change in the expression of PPARβ/δ and genes involved in glucose metabolism (*FoxO1* and *CREB*) [186]. Pesticides also induce insulin resistance by acting on insulin signaling pathways (for a review see [192]). Liver inflammation was also reported upon exposure to organophosphorus [188], OC [189], neonicotinoid [142,143] pyrethroid insecticides [144], or to triazole or imidazole fungicides [190] (Table 1).

In summary this section on the role of pesticides in oxidative stress in NAFLD shows that pesticides are pro-oxidant compounds that can induce ROS production through various mechanisms: (i) detoxification via P450-dependent processes, (ii) effects on mitochondrial respiratory complexes, and (iii) action on Sirt- and or Nrf2-dependent antioxidant responses (Figure 8). It is noteworthy that the data do not consistently come from studies performed in the liver and/or related to a metabolic impact of pesticides. Although pesticide exposure and NAFLD are not clearly correlated, the prooxidative properties of pesticides as well as their impact on various pathways involved in the regulation of metabolic homeostasis support their putative role in the development or in the progression of hepatopathy. Moreover, various experimental studies showed that pesticides could affect various pathways involved in the regulation of metabolic and hepatic homeostasis strongly suggesting their role in NAFLD. Pesticides affect oxidative stress in NAFLD, but in many studies (reported in Table 1 and Table 2), animals were exposed to high doses of pesticide and durations of exposure that are not relevant for human exposure, making any human translation of results difficult. In fact, to better decipher the role of pesticide-induced oxidative stress in the development of NAFLD, more studies are needed that link different pesticide cellular targets and/or mechanisms of action underlying ROS production with hepatic processes involved in the control of energy metabolism, such as lipid metabolism, inflammation, and gene regulation by nuclear receptors. Interestingly, nuclear receptors, such as PPARs, progesterone receptor (PR), mineralocorticoid receptor (MR), glucocorticoid receptor (GR), aryl hydrocarbon receptor (AhR), retinoic acid receptors (RARs), farnesoid X receptor (FXR), and liver X receptors (LXRs) are established or suspected targets of endocrine disrupting chemicals including diverse agrochemicals, but further analyses are needed to determine whether this can lead to the development of NAFLD/NASH [410,411].

## 6. Conclusions

Many challenges remain in understanding the etiology of NAFLD because the precise staging of its progression in the absence of liver biopsy is not yet possible. Furthermore, no therapy has been approved by the U.S. Food and Drug Administration, and lifestyle modifications (diet, exercise) are difficult to implement and adhere to, which has contributed to the failure to reduce NAFLD prevalence. In terms of implemented therapeutic interventions, vitamin E and the PPARγ ligand pioglitazone have beneficial effects on steatosis and inflammation, but they do not improve fibrosis, the strongest indicator of mortality in NAFLD/NASH. However, sustained research has led to a better understanding of the pathogenesis and progression of this condition and to the development of promising compounds to block and possibly reverse fibrosis, which are presently in clinical trials [348,412]. It is well accepted that disturbances in lipid metabolism are key to the development of NAFLD and its progression. They can affect different ROS generators, such as mitochondria, ER, and NOX [226]. Although many aspects of the contributions of ROS generators to NAFLD remain unexplored, much progress has been made in understanding how increased ROS levels trigger changes in insulin sensitivity, the activity of key enzymes of lipid metabolism, innate immune system signaling, and the level of inflammatory responses. Both macro- and micronutrients impact oxidative stress in NAFLD pathogenesis. An excess of certain macronutrients, such as cholesterol and fructose, exerts pro-oxidant effects, whereas several micronutrients have the potential to decrease NAFLD through their antioxidant properties. However, human diets are complex, and it is challenging to mimic them in experimental models. The contribution of specific dietary nutrients to oxidative stress in NAFLD is not fully elucidated and needs further research. It is noteworthy that epidemiological studies did not report a correlation between pesticide exposure and NAFLD in human. However, pesticides exposure was correlated in occupationally exposed population with metabolic disorders. Moreover, various experimental studies showed that pesticides can affect various pathways involved in the regulation of metabolic and hepatic homeostasis strongly suggesting a role in NAFLD. Pesticides affect oxidative stress in NAFLD, but in many studies (reported in Table 1 and Table 2), animals were exposed to high doses of pesticide and durations of exposure that are not relevant for human exposure, making any human translation of results difficult. In rodents, perinatal exposure to organophosphorus insecticides worsens metabolic perturbations induced by HFD [197] or by overfeeding [198,206]. The lifelong effects of perinatal exposure with respect to NAFLD/NASH development, and health in general, remain an open and crucial area of research.

## Figures and Tables

**Figure 1 biomolecules-10-01702-f001:**
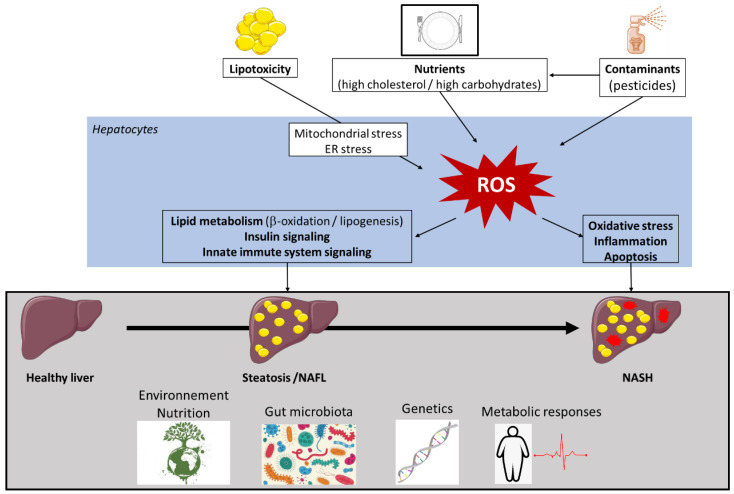
Factors involved in non-alcoholic fatty liver disease (NAFLD) pathogenesis. Multiple causes, including metabolic factors, gut microbiota, and environmental factors, operate in the context of the specific genetic background of individuals for the development of non-alcoholic fatty liver disease (NAFLD). Lipid overload induces lipotoxicity that affects different reactive oxygen species (ROS) generators, such as mitochondria, peroxisomes, and endoplasmic reticulum (ER). As environmental factors, solid and liquid foods and their contaminants contribute to ROS production. ROS generation triggers changes in insulin sensitivity, the activity of key enzymes of lipid metabolism, innate immune system signaling, inflammatory responses, and liver apoptosis. High levels of ROS enhance oxidative stress, thus creating a vicious circle. Collectively, these conditions contribute to NAFLD development and progression. Other abbreviations: NAFL, non-alcoholic fatty liver; NASH, non-alcoholic steatohepatitis.

**Figure 2 biomolecules-10-01702-f002:**
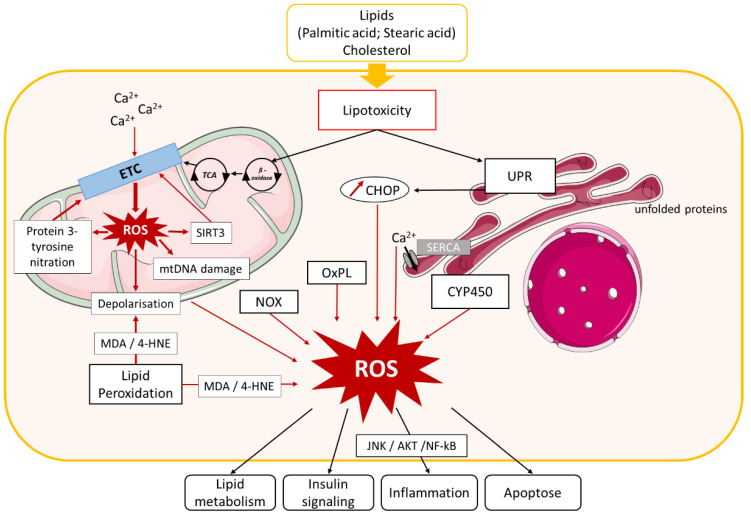
Role of oxidative stress in NAFLD. Lipid and cholesterol accumulation in hepatocytes causes lipotoxicity, resulting in activation of different pathways involved in oxidative stress. Lipid entry and alteration of mitochondrial Ca^2+^ homeostasis cause electron transport chain (ETC) dysfunction that leads to mitochondrial reactive oxygen species (ROS) production, which in turn induces Sirtuin (SIRT)3 expression, protein 3-tyrosine nitration, mitochondrial DNA damage, and membrane depolarization. SIRT3 and protein nitration also cause ETC dysfunction and exacerbate mitochondrial ROS production. Mitochondrial biogenesis dysfunction additionally induces ROS production. Lipotoxicity activates endoplasmic reticulum (ER) stress, leading to ROS production by (i) unfolded protein response (UPR) activation, which stimulates C/EBP homologous protein (CHOP) expression and (ii) Ca^2+^ homeostasis dysfunction. ROS generation in cytoplasm is induced by nicotinamide adenine dinucleotide phosphate (NADPH) oxidase (NOX) enzyme activation, oxidized phospholipids (OxPL), several cytochrome p450 enzymes (CYP450), and lipid peroxidation. ROS accumulation in hepatocytes leads to impairment of several pathways involved in NAFLD development, such as lipid metabolism, insulin signaling, inflammation, and apoptosis. Other abbreviations: mtDNA, mitochondrial DNA; TCA, tricarboxylic acid cycle; MDA, malondialdehyde; 4-HNE, 4-hydroxynonenal; SERCA, ER calcium pump sarco/endoplasmatic reticulum Ca^2+^-ATPase; JNK, c-Jun N-terminal kinase; NF-κB, nuclear factor-kappa B.

**Figure 3 biomolecules-10-01702-f003:**
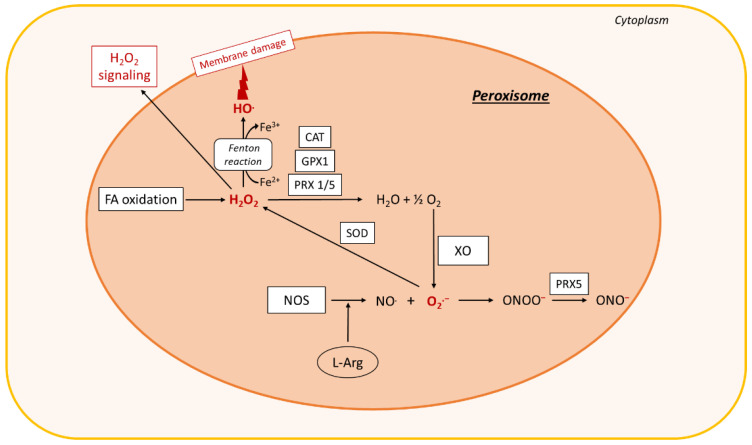
ROS production in peroxisomes. In peroxisomes, fatty acid (FA) oxidation leads to the formation of H_2_O_2_, which is decomposed by the antioxidant enzymes GPX1, CAT, or peroxiredoxin (PRX) 1 or 5, leading to the formation of O_2_ and H_2_O. O_2_ can react with XO to produce O_2_^.^^−^. Nitric oxide synthase (NOS) interaction with L-arginine (L-Arg) leads to the formation of NO^.^, which interacts with O_2_^.−^ to form ONOO^−^. ONOO^−^ is converted by Prx5 to a peroxynitrite radical ONO^.^. SOD2 dismutates O_2_^.^^−^ to H_2_O_2_. The Fenton reaction also occurs in peroxisomes and leads to membrane damage. The H_2_O_2_ is then exported to the cytoplasm, where it acts as a signaling molecule that exacerbates oxidative stress. Other abbreviations: XO, xanthine oxidase; CAT, catalase; GPX1, glutathione peroxidase 1; SOD, superoxide dismutase; Fe^2+^, ferrous iron; O_2_, oxygen; H_2_O, water; NO^.^, nitric oxide; ONOO^−^, peroxynitrite; HO^.^, hydroxyl radical; ONO^.^, peroxynitrite radical; O_2_^.^^−^, superoxide; H_2_O_2_, hydrogen peroxide.

**Figure 4 biomolecules-10-01702-f004:**
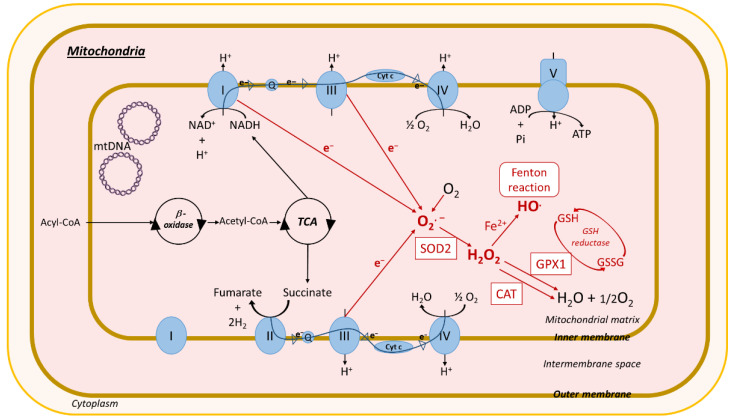
ROS production in mitochondria. Acyl-CoA imported into the mitochondria is converted by β-oxidation into acetyl-CoA, which enters the tricarboxylic acid cycle (TCA), resulting in the production of nicotinamide adenine dinucleotide (NADH) and succinate. Mitochondria have two respiratory chains composed of different complexes: (i) complexes I (NADH-ubiquinone oxidoreductase), III (cytochrome b-c1), and IV (cytochrome *c* oxidase). Complex I reduces NADH to NAD^+^ and H^+^ after electron (e−) release, and (ii) complexes II (succinate-quinone oxidoreductase), III, and IV. Complex II catalyzes the dehydrogenation and oxidation of succinic acid into furamate and 2H_2_ after e−release. Released electrons pass through complexes and other molecules in the mitochondria inner membrane ubiquinone (Q) and cytochrome c (cyt c). Complex V (F1F0-ATP synthase) uses the chemiosmotic proton gradient to power the synthesis of ATP from adenosine-diphosphate (APD) and Pi. In red: impairment of the electron transport chain results in the leakage of e− that react directly with oxygen to form the superoxide anion radical transformed in H_2_O_2_ through SOD2 activity. H_2_O_2_ can react with Fe^2+^ to form OH^.^ (Fenton reaction). H_2_O_2_ is processed into H_2_O and O_2_ by the antioxidant enzymes GPX1 and catalase. Mitochondrial ROS production causes oxidative mtDNA, lipid, and protein damage. Other abbreviations: CAT, catalase; GPX1, glutathione peroxidase 1; SOD2, superoxide dismutase; Fe^2+^, ferrous iron; O_2_, oxygen; H_2_O, water; HO^.^, hydroxyl radical; O_2_^.^^−^, superoxide; H_2_O_2_, hydrogen peroxide; ATP, adenosine-triphosphate; Pi, inorganic phosphate; NAD^+^, nicotinamide adenine dinucleotide; H^+^, hydrogen; mtDNA, mitochondrial DNA; GSH, glutathione; GSSG, oxidized glutathione; ROS, reactive oxygen species.

**Figure 5 biomolecules-10-01702-f005:**
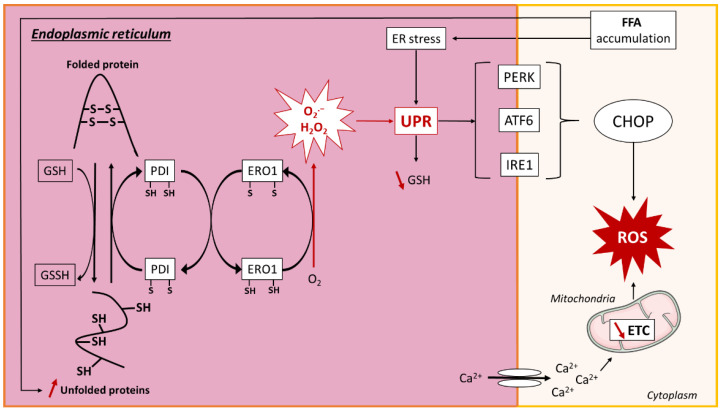
ROS production in endoplasmic reticulum. Lipid overload causes accumulation of unfolded proteins in the endoplasmic reticulum (ER) by the reduction of reduced glutathione (GSH) to oxidized glutathione (GSSH), resulting in the breakage of protein disulfide bonds (SH). The ER resident proteins, protein disulfide isomerase (PDI) and ER oxidoreductin 1 (ERO1), are involved in disulfide bond formation and superoxide (O_2_^.−^) and hydrogen peroxide (H_2_O_2_) release through electron transfer. ROS production and ER stress activate the unfolded protein response (UPR), which leads to a decrease in GSH. UPR is regulated by three transmembrane proteins: protein kinase RNA-like ER kinase (PERK), activating transcription factor 6 (ATF6), and inositol-requiring signaling protein 1 (IRE1), which promote the transcription of CCAAT/enhancer-binding protein homologous protein (*CHOP*) that further induces ROS generation. ER stress is associated with reduced sarco/endoplasmatic reticulum Ca^2+^-ATPase (SERCA) activity, leading to calcium leakage and decreased mitochondrial electron transport chain (ETC) activity, resulting in ROS production. Other abbreviations: O_2_, oxygen; FFA, free fatty acid; ROS, reactive oxygen species; Ca^2+^, calcium.

**Figure 6 biomolecules-10-01702-f006:**
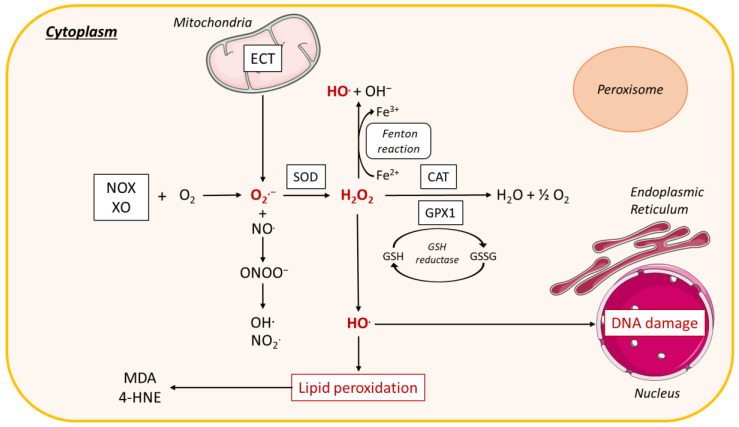
ROS production in the cytoplasm. In the cytoplasm, interaction of NADPH oxidase (NOX) and xanthine oxidase (XO) with oxygen (O_2_) leads to the formation of superoxide radical (O_2_**^.^**^−^). O_2_**^.^**^−^ dismutation by the antioxidant enzyme superoxide dismutase (SOD) forms hydrogen peroxide (H_2_O_2_), which is decomposed to O_2_ and water (H_2_O) by the enzyme catalase (CAT) or the enzyme glutathione peroxidase 1 (GPX1), leading to the oxidation of reduced GSH into glutathione disulfide (GSSG). Reactive nitrogen species (RNS) are derived from nitric oxide (^.^NO) and superoxide (O_2_**^.^****^−^**) produced via specific enzymes such as NADPH oxidase; the reaction of **^.^**NO with O_2_**^.^**^−^ produces peroxynitrite (ONOO^−^). ONOO^−^ can react with other molecules to form additional types of RNS including nitrogen dioxide (NO_2_**^.^**) as well as other types of chemically reactive free radicals (**^.^**OH). H_2_O_2_ forms a hydroxyl radical (HO**^.^**), which causes lipid peroxidation leading to malondialdehyde (MDA) and 4-hydroxynonenal (4-HNE), or interacts with DNA causing DNA damage. H_2_O_2_ can also react with ferrous iron (Fe^2+^) to form hydroxyl radical (OH^.^) and OH^−^, called the Fenton reaction. Other abbreviations: NADH, nicotinamide adenine dinucleotide; H_2_O_2_, hydrogen peroxide; DNA, Deoxyribonucleic acid; XO, xanthine oxidase; ECT, electron transport chain; H_2_O, water.

**Figure 7 biomolecules-10-01702-f007:**
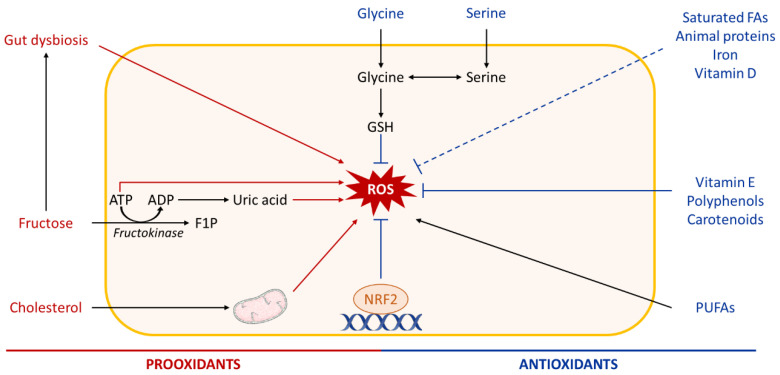
Role of nutrients in hepatic ROS production. High intake of diets rich in fructose or cholesterol promotes hepatic oxidative stress. Fructose metabolism leads to adenosine triphosphate (ATP) depletion and uric acid production, which both trigger reactive oxygen species (ROS) generation in the liver. High fructose consumption can also cause gut dysbiosis, which influences the liver oxidative status. Dietary cholesterol induces mitochondrial dysfunction and further mitochondrial ROS production. In contrast, several micronutrients such as vitamins, polyphenols, carotenoids, some amino acids (glycine, serine), and others have antioxidant properties and may be beneficial in NAFLD. Other abbreviations: ADP, adenosine diphosphate; F1P, fructose-1-phosphate; Nrf2, nuclear factor erythroid-2-related factor 2; GSH, reduced glutathione; FAs, fatty acids; PUFAs, polyunsaturated fatty acids.

**Figure 8 biomolecules-10-01702-f008:**
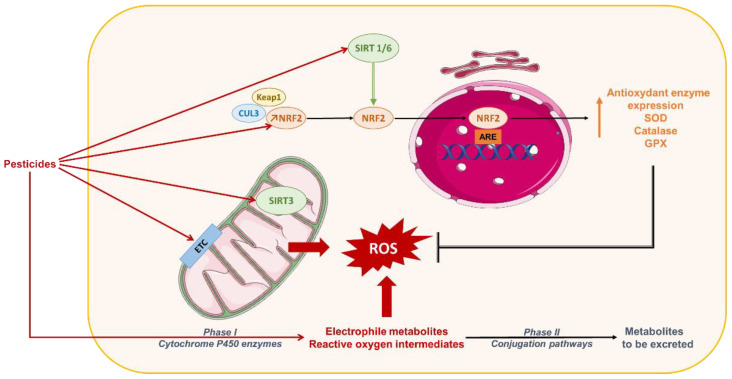
Mechanisms of pesticide-induced oxidative stress. Reactive oxygen species (ROS) can be produced from several sources during metabolism, as a result of biotransformation of pesticides by cytochrome P450 enzymes, which catalyze the oxidation reactions. Moreover, the mitochondrial complexes, except complex IV, are targets of at least one pesticide family, leading to a disruption in the electron transport chain (ETC) and ROS production. Pesticides can affect antioxidant pathways by (i) modifying the expression of erythroid2-like 2 (Nrf2), (ii) changing Nrf2 promoter methylation, and (iii) altering the expression of sirtuins (Sirts; mainly Sirt1, 3, and 6), which are members of a NAD^+^-dependent type III deacetylase enzyme family. They are considered to be stress adaptors to oxidative, genotoxic, and metabolic stress. ARE, antioxidant response element; SOD, superoxide dismutase; GPX glutathione peroxidase.

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
