# Peer review of "Oxidative Stress in NAFLD: Role of Nutrients and Food Contaminants"

_biomolecules, 2020, doi:10.3390/biom10121702_

Round 1

Reviewer 1 Report

The authors addressed all the comments of the reviewers. The manuscript is more improved compared with the first version of the article. For consistency, I suggest using the ADI (mg/kg BW/day) also in table 2 and 3. 

Author Response

W have added in the revised version of our manuscript the ADI values in tables 2 and 3. We thank reviewer for the time and effort in editing our manuscript

Reviewer 2 Report

The authors have fully responded to my comments. I consider this manuscript to be worthy of publication.

Author Response

we thank reviewer for the time and effort in reviewing our manuscript and for all the constructive comments and input to our manuscript 

Reviewer 3 Report

The resubmission of this manuscript is substantially updated and a commendable work. It is good to see some additional literature in human epidemiology that supports the hypothesis of contaminant influences in NAFLD. The limitations of the studies and important framing of the referenced work are clearly defined.

The added discussion about animal models is fantastic and helps the reader place the work in context.

My comments were addressed and additional improvements were made to the manuscript.

Author Response

We thank reviewer for the time and effort in reviewing our manuscript, for all his/her constructive comments that improve our manuscript